# Towards scientific discovery with dictionary learning: Extracting biological concepts from microscopy foundation models

Konstantin Donhauser [* † 1]   Kristina Ulicna [* 2 3]   Gemma Elyse Moran [4]   Aditya Ravuri [† 5]   Kian Kenyon-Dean [3]
Cian Eastwood [2 3]   Jason Hartford [2 3 6]

## Abstract

Sparse dictionary learning (DL) has emerged as a powerful approach to extract semantically meaningful concepts from the internals of large language models (LLMs) trained mainly in the text domain. In this work, we explore whether DL can extract meaningful concepts from less human-interpretable scientific data, such as vision foundation models trained on cell microscopy images, where limited prior knowledge exists about which high-level concepts should arise. We propose a novel combination of a sparse DL algorithm, Iterative Codebook Feature Learning (ICFL), with a PCA whitening pre-processing step derived from control data. Using this combined approach, we successfully retrieve biologically meaningful concepts, such as cell types and genetic perturbations. Moreover, we demonstrate how our method reveals subtle morphological changes arising from human-interpretable interventions, offering a promising new direction for scientific discovery via mechanistic interpretability in bioimaging.

## 1. Introduction

Large scale machine learning systems are extremely effective at generating realistic text and images. However, these models remain black boxes: it is difficult to understand how they produce such detailed reconstructions, and to what extent they encode semantic information about the target domain in their internal representations. Investigation of how models encode and use high-level, human-interpretable concepts is challenging due to the "superposition hypothesis" (Bricken et al., 2023), which states that neural networks encode many more concepts than they have neurons, and as a result, one cannot understand the model by inspecting individual neurons. One hypothesis for how neurons encode multiple concepts at once is that they are low-dimensional projections of some high-dimensional, sparse feature space. Quite surprisingly, there is now a large body of empirical evidence that supports this hypothesis in language models (Mikolov et al., 2013; Elhage et al., 2022; Park et al., 2023), games (Nanda et al., 2023) and multimodal vision models (Rao et al., 2024), by showing that high-level features are typically predictable via *linear* probing. Further, recent work has shown that model representations can be decomposed into human-interpretable concepts using dictionary learning models, estimated via sparse autoencoders (SAEs, Templeton, 2024; Rajamanoharan et al., 2024b;a; Gao et al., 2024).

However, these successes rely on some form of text supervision, either directly through next-token prediction or indirectly via contrastive objectives like CLIP (Radford et al., 2021), which align text and image representations. Further, these successes appear in domains which are naturally human-interpretable (*i.e.* text, games, natural images). As a result, one may worry that high-level features can be extracted only in settings that we already understand. This raises a natural question: can we extract similarly meaningful high-level features from completely unsupervised models in domains where we lack strong prior knowledge?

For example, in computational biology, masked autoencoders (MAE) trained on cellular microscopy images have been shown to be very effective at learning representations that recover known biological relationships (Kraus et al., 2024). However, it is not known whether analogous high-level features can be extracted from large MAE-based foundation models of cell microscopy data. These settings are precisely where extracting high-level features could be most valuable: given that models can detect subtle differences in images (even those that are very challenging for human experts to interpret), we might hope that the mechanistic interpretability techniques would enable better understanding of the ways in which these subtle differences arise.

In this work, we study the extraction of high-level features

---

[*] Equal contribution [†] Work done while interning at Valence Labs. [1]ETH Zürich [2]Valence Labs [3]Recursion [4]Rutgers University [5]University of Cambridge [6]University of Manchester. Correspondence to: Kristina Ulicna, Jason Hartford <kristina@valencelabs.com, jason@valencelabs.com>.

*Proceedings of the $42^{nd}$ International Conference on Machine Learning*, Vancouver, Canada. PMLR 267, 2025. Copyright 2025 by the author(s).

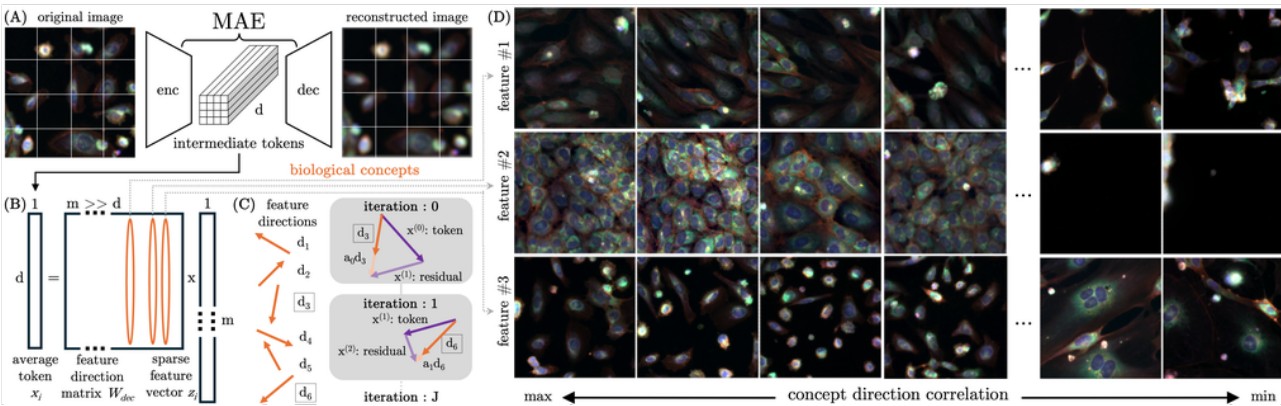

*Figure 1.* **Graphical abstract illustrating our algorithmic approach to biological concept interpretability from cell image data**, consisting of **(A)** token-wise embeddings of cell images learned by a masked autoencoder (MAE), **(B)** reconstruction of the averaged token $x_i$ through a large codebook $W_{dec}$ and sparse feature vector $z_i$ of sparsity $K$, which both get updated through **(C)** iterative codebook feature learning (ICFL) adapted from the matching pursuit algorithm for $J$ iterations. The learned codebook columns represent feature directions (orange) which correlate with **(D)** *biological concepts* visualized as cell images ranked according to the correlation strength with three selected features. Each feature captures distinct cellular morphologies: Feature #1 activates for cells with an elongated, spindle-like shape (left) and anti-correlates for sparser and clumped cells (right); Feature #2 activates for cells that are densely packed with closely arranged nuclei (left) and deactivates when cell density drops (right); and Feature #3 activates for compact, round, bright cells without cell-cell contacts almost entirely made up from just nuclei (left), in contrast to multi-nucleated cells which occupy larger areas (right).

from large-scale MAEs trained on microscopy images from a cell screening campaign containing cells that have been modulated with genetic and/or small molecule perturbations (Fay et al., 2023). Understanding the morphological changes induced by such perturbations is an inherently difficult and fundamental problem that plays a crucial role in drug discovery (Celik et al., 2024). Recent progress in this field has been made by using machine learning to learn representations of perturbations based on their corresponding cell images (Figure 1A). One direction to analyze their meaningfulness is to build relationship maps, such as by computing cosine similarities of post-processed MAE representations of cell images (Kraus et al., 2024; Celik et al., 2024; Lazar et al., 2024; Kenyon-Dean et al., 2025). However, these deep learning-based methods provide limited insights about the morphological changes arising from the perturbations: we can tell whether two perturbations are similar (or dissimilar) via cosine similarity, but we cannot tell *why* (or the ways in which) they are different. That is, we collapse the relationships between multidimensional representations down to a single score.

If we are to use dictionary learning in this context, we need to address three questions. First, *how do we extract sparse features from MAEs?* While it is possible to learn some features using TopK SAEs (Makhzani & Frey, 2013; Gao et al., 2024), we find that we achieve both better reconstruction and more selective features using an efficient variant of the matching pursuit algorithm (Mallat & Zhang, 1993). Our new proposed algorithm, Iterative Codebook Feature Learning (ICFL, Figure 1B-C), naturally avoids "dead" features

(§ 4), and is combined with PCA whitening on a control dataset which acts as a form of weak supervision (§ 5), to further improve selectivity of the extracted features (§ 6).

Second, given that MAEs are not supervised with text or any other labels or annotations, *do we learn interesting concepts at all?* We find that the DL approaches extract a wide variety of semantically meaningful features. Some are relatively simple, like light leaks or dead cells, while others capture biologically-meaningful concepts (Figures 1D & 6).

Finally, we ask *how do we evaluate the biological content of the sparse features?* We achieve such interpretations by directly involving domain experts to assess feature quality and biological relevance (Figures 1D & 6), and we introduce a carefully curated linear probing benchmark (§ 5 & § 6.2) to show that ICFL extracts features of a comparable interpretability level to biology-informed hand-crafted features.

In summary, we show how dictionary learning can be used to extract features from a large scale masked autoencoder trained on microscopy images (Kraus et al., 2024). We find features correlated with meaningful concepts, such as individual cell types or genetic perturbations. Moreover, via linear probing, we show that the learned features preserve significant amounts of biologically meaningful information.

## 2. Related work

**Dictionary Learning** The dictionary learning (DL) problem became popular in the 1990's (Mallat & Zhang, 1993;

| | w/o PCA whitening | w/ PCA whitening |
|---|---|---|
| **ICFL** | 55 | 341 |
| **TopK** | 7640 | 8026 |

*Table 1.* The number of "dead features" (out of 8192) activated less than a fraction of $10^{-5} \times$ during the last 1000 training steps, for both TopK and ICFL with and without PCA whitening (§ 5).

Olshausen & Field, 1996) and has since been extensively studied in the literature (*e.g.*, (Aharon et al., 2006; Donoho et al., 2001; Spielman et al., 2012)). Recently, (Bricken et al., 2023) proposed to use DL to extract features from internal representations of LLMs, with several follow-up works (*e.g.*, (Rajamanoharan et al., 2024b;a; Gao et al., 2024)), including different modalities such as vision (*e.g.*, (Gorton, 2024; Rao et al., 2024)). These works build upon the assumption that large-scale transformer models encode "concepts" as linear directions (as reviewed in (Jiang et al., 2024)), commonly referred to as the "linear representation hypothesis". Rajendran et al. (2024) provides the theoretical conditions under which "concepts" can be provably recovered from data. Finally, similar approaches have been previously employed directly on raw data, such as for correcting unwanted variation in gene expression data (Jacob et al., 2016) or for detecting and counting objects by learning templates in holographic images (Yellin et al., 2017).

**Causal representation learning and explainability** The disentanglement and causal representation literature (CRL) share the goal of learning high-level, interpretable concepts from data (Bengio et al., 2013; Kulkarni et al., 2015; Higgins et al., 2017; Chen et al., 2016; Eastwood & Williams, 2018; Schölkopf et al., 2021). Two key differences with the dictionary learning approach are: (i) disentanglement/CRL methods consider low-dimensional representations to capture the factors of variation in data, whereas overcomplete dictionary learning seeks a higher-dimensional representation to capture a large set of sparsely-firing concepts; and (ii) disentanglement/CRL methods aim to be inherently interpretable, whereas this paper considers a post-hoc approach to interpret pre-trained models. Related work on post-hoc explainability also learns "concept vectors" in neural network internal states (Kim et al., 2018; Ghorbani et al., 2019); a key difference is that these methods use class-labeled data, whereas this paper uses an unsupervised approach to discover concepts. Another line of feature-visualization works aim to interpret internal states/neurons by finding the data points (or gradient-optimized inputs) that lead to maximal activation (Mordvintsev et al., 2015; Olah et al., 2017; Borowski et al., 2021). In contrast to these local data point centric approaches, dictionary learning methods seek global parameters (dictionaries) to explain models.

## 3. Background

**The superposition hypothesis** Let $x \in \mathbb{R}^d$ denote a representation obtained from a transformer model; as an example, $x$ may be the (aggregated) embedding of a token (resp. tokens) after the first half of the transformer layers. Bricken et al. (2023) hypothesize that (i) such token representations $x \in \mathbb{R}^d$ are linear combinations of concepts; (ii) the number of available concepts $M$ significantly exceed the dimension of the representation $d$; and (iii) each token representation is the sum of a sparse set of concepts. These desiderata are satisfied by the following model that is widely studied in compressed sensing and dictionary learning:

$$x \approx Wz = \sum_{m=1}^{M} z_m W_m \qquad \text{where } \|z\|_0 \ll d \quad (1)$$

where $W \in \mathbb{R}^{d \times M}$ is a latent dictionary matrix and $z \in \mathbb{R}^M$ is a sparse latent vector. In this paper, we will refer to the columns $W_m$ as "feature directions" and $z$ as "features".

**Feature learning using TopK SAEs.** Given a set of representations $\{x\}_{i=1}^N$, learning both $W$ and $\{z\}_{i=1}^N$ is a *dictionary learning* or *sparse coding* problem (Olshausen & Field, 1997), with a long history of works proposing efficient algorithms with provable guarantees (Aharon et al., 2006; Arora et al., 2014; 2015). In the context of mechanistic interpretability, the dominant choice for learning these parameters are two-layer sparse autoencoders (SAEs). In this paper, we compare to the state-of-the-art method called TopK SAE, originally proposed by (Makhzani & Frey, 2013) and recently studied by (Gao et al., 2024). Following their notation, the model for tokens $i \in [N]$ is:

$$x_i = W_{\text{dec}} z_i + b_{\text{pre}}, \quad \text{with } z_i = \text{TopK}(W_{\text{enc}} x_i - b_{\text{pre}}),$$

where $\text{TopK}(\cdot)$ is an operator that sets all but the $K$ largest elements to zero. The parameters $\{W_{\text{dec}}, W_{\text{enc}}, b_{\text{pre}}\}$ are

---

**Algorithm 1** Batched-OMP

1: **Input:** Parameters $W_{\text{dec}}, b_{\text{pre}}$; model representation $x$; # sparse features $L$ per iteration and # iterations $J$
2: Initialize $x^{(1)} := x - b_{\text{pre}}$
3: **for** $t = 1$ TO $J$ **do**
4:     Select top $L$ columns of $W_{\text{dec}}$ which maximize $\langle W_{\text{dec},m}, x^{(t)} \rangle$
5:     Solve $z^{(t)} = \text{argmin}_z \|x^{(t)} - W_{\text{dec}} z\|_2^2$ with $z$ non-zero only for $L$ selected columns
6:     Update $x^{(t+1)} := x^{(t)} - W_{\text{dec}} z^{(t)}$
7: **end for**
8: **Output:** Sparse features $z := \sum_{t=1}^{J} z^{(t)}$

---

learned by minimizing the reconstruction loss:

$$L(W_{\text{dec}}, b_{\text{pre}}) := \sum_{i=1}^{N} \|x_i - \widehat{x}_i\|_2^2, \quad \text{where} \quad (2)$$

$$\widehat{x}_i = W_{\text{dec}}\text{TopK}(W_{\text{enc}}x_i - b_{\text{pre}}) + b_{\text{pre}}. \quad (3)$$

A problem with the above optimization is that some feature directions $W_{\text{dec},m}$ are barely used; that is, we have inactive features $z_{im} = 0$ for almost all $i \in [N]$. This is called the "dead feature" phenomenon. To reduce the amount of dead features, (Gao et al., 2024) introduce an additional reconstruction error term containing only these feature directions to encourage their usage in the reconstruction (see Table 1).

**Orthogonal Matching Pursuit**    The OMP algorithm (Mallat & Zhang, 1993) is widely used in signal processing applications. It assumes that the dictionary, $W$, is given, and after initializing $x^{(0)} = x$ it iteratively finds a $k$-sparse vector $\hat{z}$ by repeating the following steps:

1. Find the column in $W$ that solves $w_i = \arg\max_i |W_i^\top x^{(t)}|$ and append it to the matrix $W_s$ of selected columns.

2. Solve $z^{(t)} = \arg\min_z \|x^{(t)} - W_s z\|^2$

3. Update $x^{(t+1)} := x^{(t)} - W_s z^{(t)}$

and terminate when either $x^{(t+1)}$ is sufficiently small (i.e., the algorithm has converged) or $z^{(t)}$ is $k$-sparse for some pre-specified $k$.

## 4. Iterative codebook feature learning (ICFL)

We can interpret the OMP algorithm as playing the same role as the encoder in TopK SAEs in that the algorithm provides a mapping from a representation, $x$, to $z$ such that $z$ is (at most) $k-$sparse. But standard OMP is not useful for dictionary learning because (i) the dictionary is pre-specified and fixed for the duration of the algorithm and (ii) its sequential operation is slow for large $k$. With two simple modifications we address these issues in an algorithm we call Iterative Codebook Feature Learning (ICFL), which computes the features $z$ using a *batched* variant of MP. During training, we then update the feature directions, $W_{\text{dec}}$, using gradient descent.

**Computing the features $z$ using batched-OMP**    Given the current matrix of feature directions $W_{\text{dec}}$, we first select a batch of the top-$L$ columns most aligned with $x^{(1)} = x$. Then, we proceed as before, learning the features $z^{(1)}$ that best reconstruct $x \approx W_{\text{dec}}z^{(1)}$, using only these columns (i.e. $z^{(1)}$ is $L$-sparse). Next, to obtain $z^{(2)}$, we repeat this step, but replace $x$ with the residual $x^{(2)} = x - W_{\text{dec}}z^{(1)}$.

Repeating this process, the final output $z$ is taken to be $z = \sum_{t=1}^{J} z^{(t)}$. Consequently, $z$ is at most $k = JL$-sparse. The key advantage over SAEs is that early iterations subtract dominant feature directions from $x$, allowing the algorithm in later iterations to select a broader set of feature directions that are not as correlated with the main features in $x$.

**Learning the feature directions using gradient descent**
Given the features $z$, we then update the decoder parameters $\{W_{\text{dec}}, b_{\text{pre}}\}$. We do so via batched gradient descent minimizing the reconstruction loss:

$$L_{\text{ICFL}}(W_{\text{dec}}, b) := \sum_i \|x_i - W_{\text{dec}}z_i - b_{\text{pre}}\|_2^2. \quad (4)$$

As $z_i$ is fixed in this gradient step, the algorithm does not need to propagate gradients through $z$ in order to learn an encoder; instead the "encoder" is simply batched-OMP which will use any feature direction $W_{\text{dec},m}$ that aligns with some $x_i$, and hence as long as some $x_i$ aligns with $W_{\text{dec},m}$, that $W_{\text{dec},m}$ will receive gradient updates. We find that this procedure is much more robust against the emergence of "dead" features during training, as shown in Table 1.

In practice, we initialize $W_{\text{dec}}$ using the standard *pytorch* initialization for a linear transform and leverage random resets to ensure that the columns of $W_{\text{dec}}$ are not too correlated (high correlation is sometimes referred to as feature collapse). Specifically, after every 100 stochastic gradient descent steps, we take every pair of columns of $W_{\text{dec}}$ that have cosine-similarity above 0.9 and randomly initialize one of the pairs with a vector selected uniformly at random from the hypersphere. One could think of the random resets as playing a role analogous to the projection step in projected gradient descent, in order to enforce the constraint that $\text{cosim}(W_i, W_j) < 1 - \delta$. However, unlike projected gradient, we are not deterministically projecting onto the boundary of the constraint set; instead, the random reset ensures that we are at some (random) point in the interior of the constraint set (with high probability). This trades off the compute costs and algorithmic complexity of a true projection step for a cheap alternative that works well in practice. Before running Algorithm 1, we always center the representations $x$ by subtracting the average representation of unperturbed samples from the control distribution, such that the origin represents the unperturbed state. Before ICFL, we normalize the representations to have unit norm.

### 4.1. PCA whitening using a control dataset

As dictionary learners seek to minimize the Euclidean distance between the model representations $x$ and their reconstructions $\hat{x} = W_{\text{dec}}z$, the learned features $z$ are naturally biased towards capturing the dominant directions in the data (i.e., those that explain the most variance). In biological applications, these directions will typically correspond bio-

logical variation that is present in all experiments (e.g. morphological changes induced by the cell cycle), and hence are not the most useful concepts to learn. To avoid this variation dominating the reconstruction loss, we use a dataset of control samples as a form of weak supervision, downweighting dominant directions in this control dataset as we know they do not correspond to the biological perturbations of interest. In particular, let $\mathbf{X}$ matrix of representations of $n$ samples from the distribution of images of unperturbed / control cells, $P(X|\emptyset)$. Let $\mathbf{T} : \mathbf{X} \to W(\mathbf{X} - \mu)$ denote the linear map the maps the control cells to a distribution with zero mean and unit covariance, where $W$ is a PCA matrix. By applying the linear tranformation, $T$, to the peturbed cells sampled from, $P(X|\delta_i)$ (for all perturbations $\delta_i$) *before normalization*, we effectively scale up the differences between perturbation representations. For our multi-cell data, unperturbed HUVEC-cell images act as our control dataset. Note that similar PCA whitening on a control dataset has been used to improve the quality of the learned multi-cell image representations (Kraus et al., 2024; Kenyon-Dean et al., 2025).

## 5. Experimental setup

**Data source and foundation model** We evaluated our DL approach on two large-scale masked autoencoders trained on cellular microscopy Cell Painting image data using $256 \times 256 \times 6$ pixel crops as input and a patch size of $8 \times 8$, following the same procedures as described in Kraus et al. (2024); Kenyon-Dean et al. (2025). These models were trained on data from multiple cell types that were perturbed with CRISPR gene knockout perturbations. Both models used the architecture hyperparameters from Kraus et al. (2024); Kenyon-Dean et al. (2025), with the smaller of the two using the ViT-L/8 configuration, while the larger model used the ViT-G/8 configuration. We refer to these models as *MAE-L* and *MAE-G*, respectively. We obtain a single token per input crop by aggregating all patch tokens (excluding the class token). For both the residual stream and the attention output (after the out-projection), the dimension $d$ of

| Task | Classes | Samples | BTA | Threshold | |
| | | | | 0.5 | 0.2 |
|---|---|---|---|---|---|
| *Cell type* | 23 | 110,971 | 97.2% | 73 | 455 |
| *Batch* | 272 | 80,000 | 87.8% | 11 | 77 |
| *siRNA* | 1,138 | 81,224 | 51.6% | 0 | 141 |
| *CRISPR* | 5 | 79,555 | 94.6% | 0 | 2 |
| *Group* | 39 | 57,863 | 32.1% | 0 | 37 |

*Table 2.* Five tasks (§ 5) classified by linear probes trained on well-level aggregated representations from the residual stream from an intermediate layer from *MAE-G* (left) and counted for features with average selectivity above threshold for at least one label (right).

the tokens (representations) are 1024 and 1664 for MAE-L and MAE-G, respectively. All the visualizations used Cell Painting microscopy images from the RxRx1 (Sypetkowski et al., 2023) and RxRx3 (Fay et al., 2023) datasets.

We extract the tokens from layers 16 (*MAE-L*) and 33 (*MAE-G*), respectively. The motivation for using intermediate instead of final layers is that these tokens are more likely to capture abstract high level concepts that are used by the model *internally* to solve the self-supervised learning task (Alkin et al., 2024). We selected this layer by finding the maximized linear probing performance on the functional group task from the original embeddings (§ 5).

**Preserving linear probing signals** To investigate whether the features found by sparse DL retain important information from the original representation, we define five different classification tasks (Table 2). For each classification task, we use a separate (potentially overlapping) dataset and split it into train and test data to distinguish the labels across:

(1) 23 different cell types which are almost perfectly distinguishable via linear classification.

(2) 272 different experiment batches. Even in controlled conditions, subtle changes in experimental conditions can induce strong *batch effects*, *i.e.* changes in experimental outcomes due to experiment-specific variations unrelated to the perturbation that is being tested.

(3) 1138 siRNA perturbations from the RxRx1 dataset (Sypetkowski et al., 2023), where the single-gene expression is partially (or completely) silenced using short interfering (si-)RNA, which targets the gene mRNA for destruction via the RNA interference pathway (Fire et al., 1998). As the extent of siRNA knockdowns is hard to quantify and prone to significant but consistent off-target effects, we also evaluated:

(4) 5 single-gene CRISPR perturbation knockouts which induce strong and consistent morphological profiles across cell types, known as "perturbation signal benchmarks" (Celik et al., 2024). Unlike the siRNA approach, CRISPR (Jinek et al., 2012) cuts the gene DNA directly, which induces a mutation and represses the gene function. As perturbations to similar genes often result in similar visual effect, we also established:

(5) 39 functional gene groups composed of CRISPR single-gene knockouts categorized by phenotypic relationships between the genes, including major protein complexes, as well as metabolic and signaling pathways. Each gene group targets similar or related cellular process, which results in inducing morphologically similar changes in the cells (Celik et al., 2024).

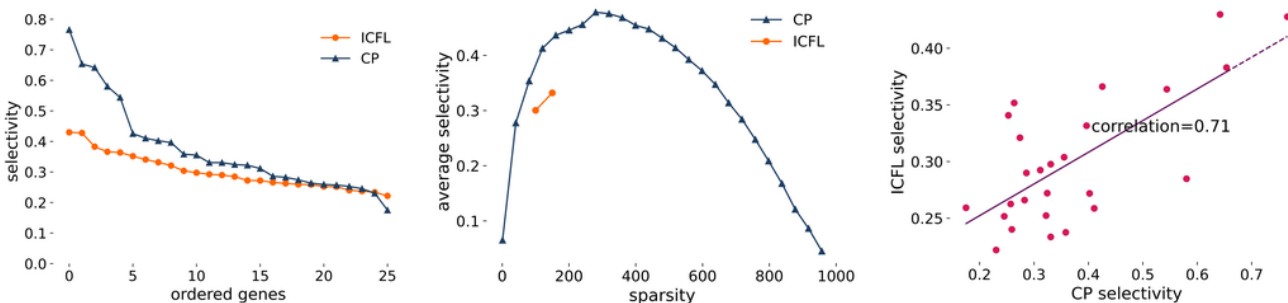

*Figure 2.* **Comparison of feature average selectivity scores** from *CellProfiler* (CP) and ICFL for a subset of Task (3). **(A)** Max avg selectivity scores for each label in descending order, **(B)** averaged across labels at different thresholds for CP and sparsity levels for ICFL, as a function of the average number of non-zero values. **(C)** Correlation of max avg selectivity scores for each label between CP and ICFL.

To remove the impact of spurious correlations between perturbations and batch effects on the test accuracy, we always use mutually exclusive experiments for test and train data, except for Task (2), where the goal is to predict the experiment. Except for Task (1), all classification tasks use HUVEC cells. The representations are always well-level aggregated, *i.e.* we take the mean over the total of 1,024 ($8 \times 8$ pixel regions) individual 1,024-dimensional tokens from all $36\ 256 \times 256$ non-edge crops from an $2,048 \times 2,048$ pixel image of a given well (Celik et al., 2024). Due to heavy class imbalance (particularly for Task (1)), we train our linear probes using logistic regression on a class-balanced cross-entropy loss and report *balanced test accuracy* (BTA).

**Training the DL models** By default, we always choose a sparsity of $K = 100$ for TopK SAEs and $J = 20, L = 5$ (resulting in a max sparsity of 100) for ICFL as described in § 4, and use a total of $M = 8192$ features. Unless otherwise specified, we always apply the PCA whitening described in § 4.1 and use representations from the residual stream. We train the models using 40M tokens (one token per image crop) with a batch size of 8192 for 300k iterations. Our learning rate is $5 \times 10^{-5}$ for all experiments. Similar to Gao et al. (2024), we observed that changing the learning rate has a limited impact on the outcome. We present an ablation for the learning rate in Appendix C.

## 6. Experimental results

In this section, we present our results using features extracted from ICFL combined with PCA whitening (if not further specified), and comparing with Top-K SAEs (§ 6.3).

### 6.1. Features are correlated with biological concepts

Initially, we investigate how strongly correlated the features are with labels from the classification tasks (Table 2).

**Selectivity of features for biological concepts** For each dataset associated with a classification task, we extract from every image a feature vector using the center crop as input to the MAE. For each feature, we then compute two selectivity scores: the average, or **avg selectivity** score, which is the % of times that the feature is active given that label $i$ occurs minus the % of times the feature is active given any other label. As a stronger notion of correlation, we also use the maximum, or **max selectivity** score, that subtracts the maximum % for any other label. The selectivity score has been originally proposed in the context of neuroscience (Hubel & Wiesel, 1968) and has also been used by (Madan et al., 2022) to measure the "monosemanticity" of neurons.

In Table 2, we present the number of features exhibiting an average selectivity greater than a given threshold for at least one label, across all five classification tasks. We observe that dominant concepts, such as cell types, batch effects, and siRNA perturbations that induce strong morphological changes, comprise a substantial portion of features displaying high selectivity. Moreover, for labels from the functional gene groups in Task (5), we identify more than 100 features with selectivity scores of at least 0.1.

**Separation along feature directions** The selectivity score analysis showed that activation patterns of the sparse features can be strongly correlated with genetic perturbations. We now consider the feature directions, and show that certain feature directions can distinguish specific genetic perturbations. Specifically, in Figure 3, in each histogram we plot the cosine similarity of a feature direction with both (i) representations from a specific siRNA perturbation (blue) and (ii) representations from all other perturbations (orange). The plot shows that certain feature directions effectively sep-

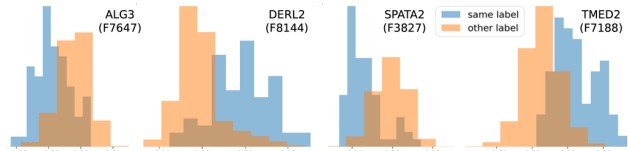

*Figure 3.* **Cosine similarity histograms for selected pairs of representations** from Task (3) and features directions, if its associated perturbation is applied (blue) *vs.* any other perturbation (orange).

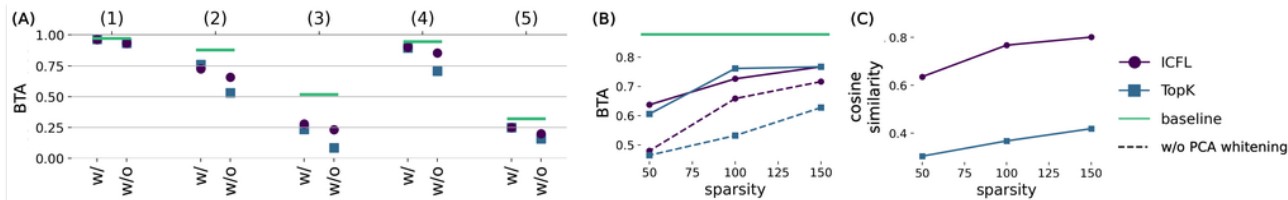

*Figure 4.* **Comparison of ICFL with TopK SAEs. (A)** Balanced test accuracy (BTA) of linear probes trained on the original representation (solid line) and reconstructions from ICFL and TopK SAEs with and without PCA whitening for five tasks from § 5. **(B)** Balanced test accuracy (BTA) as a function of the sparsity (dashed line is the original representation $x$) for classification Task (5). **(C)** Cosine similarity of reconstruction and original representations as a function of sparsity for tokens from a hold-out validation dataset.

arate the two groups, and as such these feature directions capture important biological information.

**Linear probing for biological signals** Finally, as a measure for how much "biologically-relevant" information is lost when extracting sparse features, we compare the accuracy of linear probes on the representations, $x$, with linear probes on the reconstructions from ICFL, $\widehat{x} = \widehat{W}_{\text{dec}}\widehat{z}$. Almost the entire signal is preserved for simple concepts such as cell types, batch effects and perturbations with strong morphological changes (1,2,4) (Figure 4A). For the difficult tasks of distinguishing between many genetic perturbations (3,5), a substantial amount of the linear signal is preserved.

### 6.2. Comparison with features from *CellProfiler*

We compared the average selectivity scores of features from ICFL with those from a set of 964 handcrafted features generated by *CellProfiler* (CP) (Carpenter et al., 2006). These features were designed by domain experts and are widely used for microscopy image analysis. In this experiment we consider a subset of images from the public RxRx1-dataset (Sypetkowski et al., 2023). CP extracts features for each cell; for a multi-cell image, we calculate an image-level CP feature by averaging the CP features from each cell in the image. The CP features are not sparse; to make them comparable with the sparse ICFL features, we thresholded the CP features at $\alpha$ and $1 - \alpha$ quantiles, with $\alpha$ chosen such that the average number of non-zeros was $\approx 100$. A CP feature was considered to be "activated" when its value, under perturbation conditions, exceeded these quantiles.

In Figure 2A, we plotted the highest average selectivity score for each genetic perturbation, a subset of Task (3). We found that the features extracted by ICFL almost matched the selectivity scores of the handcrafted, human-designed features. Additionally, in Figure 2B, shows the average score across all labels as a function of various thresholding levels for the CP features. We again observed that our features performed comparably to CP features. Interestingly, CP features peaked at high levels of non-zeros ($\approx 300$), leaving future work to assess whether this peak selectivity could be matched using deep learning-based approaches which used significantly fewer non-zero elements. In Fig-

ure 2C we illustrate the correlation between the best average selectivity scores from the CP and ICFL features for each label. We found a strong correlation between the two sets of features (Pearson coefficient of $0.71$), suggesting that ICFL was capable of identifying features that captured patterns similar to those detected by CP.

### 6.3. Comparison of ICFL with TopK SAE

Finally, we found that ICFL, when combined with PCA whitening, retained more biological signal than TopK SAEs. We first evaluated the reconstruction of the respective unsupervised DL algorithms by comparing the cosine similarity between the original and reconstructed representations. Figure 4C shows that the reconstruction quality of ICFL was much higher than TopK SAE for the same sparsity constraints when using PCA whitening. We provide further ablations in Appendix C.

To evaluate biologically meaningful information retention, Figure 4A compares the linear probing accuracies for the two algorithms. Both TopK SAE and ICFL features yielded a similar linear probing accuracy, while we could see a clear drop if no PCA whitening was used during pre-processing. Figure 4B shows that while we found that increasing the number of non-zeros improved the average balanced test accuracy, the effect was not as strong as PCA whitening.

Finally, in Figure 8 we show the selectivity scores for both ICFL features and TopK SAEs. We saw that ICFL features consistently achieved higher selectivity scores than TopK SAE features. Moreover, especially for cell types, we observed a high *max* selectivity across almost all cell types, while for more complex features we observed a more modest selectivity score of more than $\approx 0.1$ across all labels.

## 7. Interpretability analysis of cell imaging data

In this section, we illustrate non-trivial patterns captured by selected features, exemplifying how domain experts can interpret and validate biological concepts learned by DL. To understand what parts of each image are most "aligned" with a particular feature, we generate heatmaps for each image by calculating the cosine similarity of the feature direction with each of the $8 \times 8$ image patch token representations.

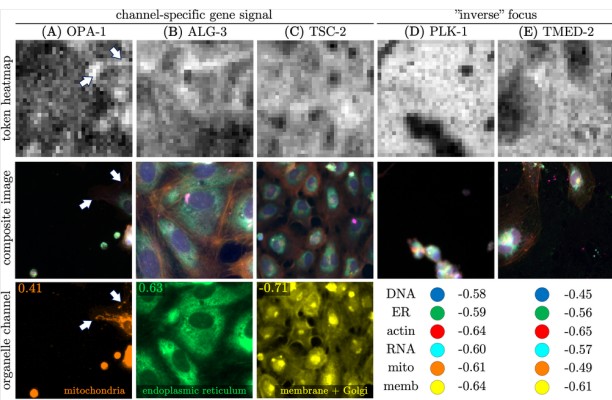

*Figure 5.* **Channel-specific features visualization** from selected single-gene perturbations. Token heatmaps (top) are plotted above the composite 6-channel images (middle) and channel-specific staining images (bottom) of selected subcellular compartments, along with the channel-heatmap Pearson correlation coefficients.

## 7.1. Channel-specific perturbation signal

We begin by examining the extent to which we can recover channel-specific signal associated with single-gene perturbations. For this exercise, we queried 3 specific single gene perturbations: (i) OPA-1, which contributes to the maintenance of correct shape of mitochondria, (ii) ALG-3, which aids in the modification of proteins and lipids in the endoplasmic reticulum (ER) after synthesis, and (iii) TSC-2, which contributes to the control of the cell size (Figure 5).

**OPA-1** The mitochondrial channel shows that most correlated tokens are overlaid with distant regions where enlarged mitochondria are present (Figure 5A; white arrows). Qualitative channel examination highlights this delicate detail which is not obvious from the composite images, confirming that our approach identifies channel-specific level of detail.

**ALG-3** The most aligned tokens appear specific to regions of endoplasmic reticulum (ER) and RNA with which ALG-3 co-localizes. This dense image suggests that our token heatmap is prevalently focused on ER-specific information (Figure 5B), which is consistent with ALG-3 function in aiding the attachment of sugar-like groups to proteins.

**TSC-2** We examine the plasma membrane- and Golgi apparatus-specific channel to relate perturbed cell size control to the token alignment. We confirm that this channel correlates most strongly with the queried concept direction, but this time in a negative direction. As the plasma membrane — and, hence, cytoplasmic area — are the most extensive from the cell center, the mostly aligned tokens appear to focus specifically on regions which are *not* covered by the cell membrane Figure 5C. Although this behavior is harder to interpret, it is suggestive of that the salient feature for these perturbations is the *lack* of cell density in a well.

**Inverse focus** Monitoring of the lack of channel-specific signal is a plausible explanation in other images where the tokens are not always co-localized with regions occupied by cells. Several genes appear to follow an "inverse" trend, including PLK-1, which enables cell cycle progression through mitosis, and TMED-2, which helps to regulate intracellular protein transport. While both of these gene perturbations render the cells in a characteristic affected state (small, clumped cells struggling to divide *vs.* large, spread out, actively dividing cells), their most aligned tokens correspond to areas *not* covered by cells, confirmed by highly negative correlations across all channels (Figure 5D-E).

## 7.2. Interpretability analysis on single-cell level

We continue by focusing on a single feature which demonstrated a clear biological relationship. This feature is strongly correlated with gene knockouts from the *adherens junctions* pathway, a label from the functional gene perturbation group from Task (5) (§ 5). The *adherens junctions* are simple to visually interpret as their function is to connect cell membranes to cytoskeletal elements and form cell-cell adhesions; they can be thought of as "glue proteins" that stick cells together to maintain tissue integrity.

The images most correlated with this feature direction reflect this disrupted cellular morphology; the images mostly comprise of small, bright and isolated cells (Figure 6A-B). Note that CRISPR genetic perturbation are not perfect and affect most but not all cells, leaving some cells in an unperturbed, control-like state. In the case of *adherens junctions*, the perturbation-affected cells would appear round, bright and isolated, while the cells which seem to have escaped the perturbation would appear flat, large and wide-spread, attempting to establish connections with their neighbors, which we successfully confirm (Figure 6A-B, white arrows).

Strikingly, our token-level heatmaps appear to have recognized the difference between these two cell populations, as the tokens corresponding to control-like cells are *not* aligned with the feature direction of the image (Figure 6C-D, dark regions). These "dark" areas consistently belong to cells whose actin meshwork extensively protrudes away from the cell center, representative of control-like cells. It appears that these dark regions are misaligned with the concept direction because these cells do not actively contribute to the representation of the perturbation; instead, they represent noisy signal which is present across almost every image. On the other hand, the regions *most* correlated with the feature direction belong to areas surrounding the perturbed, compact cells and appear to form a ring-like pattern around them. As the perturbation results in smaller and rounder cells, this suggests that the concept corresponds to the expected but missing actin (Figure 6B-D, white circles).

Next, we aimed to quantitatively confirm whether the token-

level heatmaps enable a consistent separation between the control-like cells and correctly perturbated cells. To do so, we selected 5 cell images from the *adherens junctions* pathway (further shown in Figure 10), in which a domain expert first manually counted the cells, classified each cell instance as (un-)perturbed, and manually segmented them. The corresponding token-level heatmaps were then examined to annotate cell instances by whether they align with the overall feature direction ('bright') or not ('dark').

Out of 31 cells which were scored as control-like by the human annotator, the token level heatmap areas corresponding to these single cells were "dark" in 27 instances (Table 3). The darker cell areas correspond to tokens with lower alignment to the general concept direction of the image, and hence are indicative of lower importance of these areas in the overall image for the functional group classification task. Overall, we report that the token heatmaps are capable of recalling 87% of the manually labelled cells by an expert annotator (Table 3), hinting at the potential of our approach to separate cell subpopulations in a supervision-free manner.

To remove the human subjectiveness of what constitutes a 'dark' *vs.* 'bright' cell in the token heatmap, we continued to evaluate this cell identity scoring approach in an unbiased yet quantitative way. To this end, we generated pixel-level segmentation masks of 3 categories: background, perturbed cells and unaffected cells (Figure 6C). The histograms of the relative token alignment per cell identity confirm a statistically significant difference between the control-like and perturbed single-cell instances, which are equally informative of the perturbation as the background (Figure 6E). As typical control images are more dense (as in Figure 1D, feature #2 left), the exposure of image background is an important characteristic of the *adherens junctions* phenotype.

In biological systems, complex mixtures of cell morphologies are frequently observed; however, identifying which exact cell instance belongs to which population would require deep expertise and time-consuming manual labeling at scale.

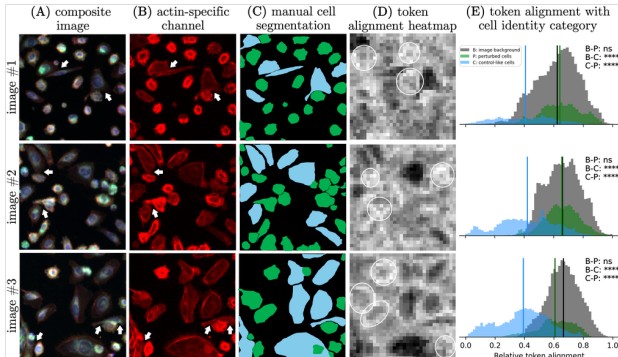

*Figure 6.* **Interpretation of single-cell identities from cell images** through **(A)** composite images and **(B)** their actin-specific channel, which strongly correlates with a feature from the *adherens junctions* gene group. **(C)** Segmented single cells classified based on morphology and interactions with neighbors (white arrows). Token-level alignment as **(D)** heatmaps highlighting cells with ring-like boundaries (white circles) and **(E)** category histograms with compared distributions (means as solid lines, one-sided Mann Whitney U test; *ns*, no significant difference; ****, $p < 0.0001$).

Tools capable of accurately and effectively disentangling these mixtures with single-cell precision in an unsupervised manner therefore remain scarce. Our token-level analysis presents a potential avenue for enabling modern approaches to scientific discovery, leveraging methods from mechanistic interpretability to uncover previously unknown biological concepts at the resolution of individual cells.

## 8. Conclusion

In this paper, we have explored the extent to which dictionary learning (DL) can be used to extract biologically meaningful concepts from microscopy foundation models. The results are encouraging: with the right approach, we were able to extract sparse features associated with distinct and biologically-interpretable morphological traits. That said, these sparse features are clearly incomplete: their linear probing performance significantly drops on tasks that involve more subtle changes in cell morphology. It is not clear to what extent this limitation stems from our current DL techniques, the scale of our models, or whether these more subtle changes are simply not represented linearly in embedding space. Nonetheless, it is evident that the choice of the DL algorithm matters to extract meaningful features.

We also proposed a new DL algorithm, Iterative Codebook Feature Learning (ICFL), and the use of PCA whitening on a control dataset as a form of weak supervision for the feature extraction. In our experiments, we found that both ICFL and PCA significantly improve the selectivity of extracted features, compared to TopK sparse autoencoders. We hope that future work further explores the use of DL and other mechanistic interpretability techniques for scientific discovery, and the use of ICFL for other modalities like text.

| Image | Manual Labeling | | | ICFL Recall | |
|---|---|---|---|---|---|
| *Cells:* | *Total* | *Perturbed* | *Control* | *'Control'* | *[%]* |
| A | 28 | 22 | 6 | 6 | 100.0 |
| B | 34 | 26 | 8 | 6 | 75.0 |
| C | 20 | 17 | 3 | 3 | 100.0 |
| D | 23 | 16 | 7 | 6 | 85.7 |
| E | 19 | 12 | 7 | 6 | 85.7 |
| **Total:** | **124** | **93** | **31** | **27** | **87.1** |

*Table 3.* **Tokens align based on single-cell identities in multi-cell images.** 'Control' recall is calculated as percentage of control-like ('dark') cells in heatmaps over all human-labeled cells. Note that 'perturbation' recall is 100% in all images from Figure 10.

## Impact Statement

This is an exploratory paper in the field of AI-driven scientific discovery. This paper primarily motivates research on the use of mechanistic interpretability for scientific discovery. The findings support the observation that deep networks are able to learn interesting semantically meaningful features. If future iterations of this work could reliably elucidate the mode of action of perturbations on cell, that would enable a dramatic improvement of our understanding of cell biology; but to get there, we need more reliable mechanistic interpretability methods of disentangling representations into biologically meaningful features.

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

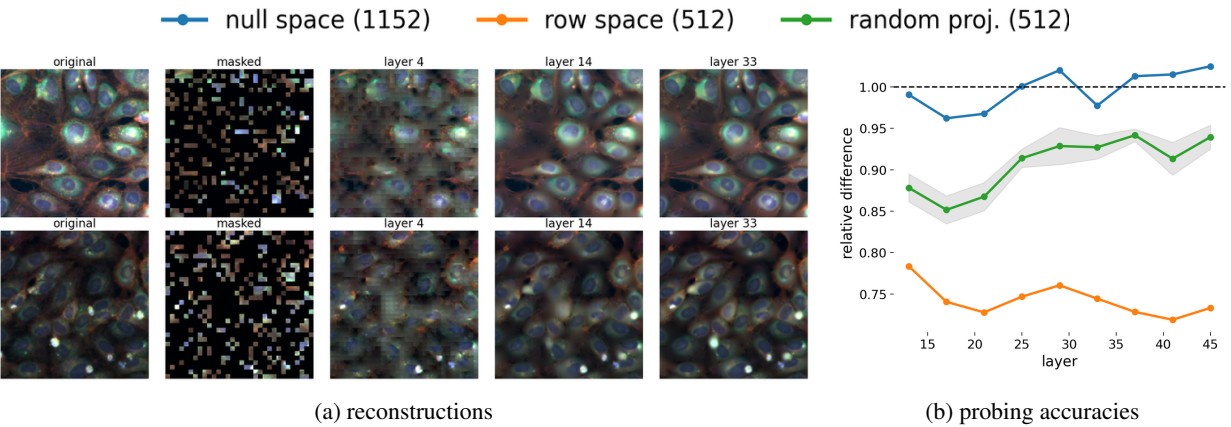

(a) reconstructions                    (b) probing accuracies

*Figure 7.* **Decoding of tokens from intermediate transformer layers.** (A) Sample reconstructed images when decoding from 3 different intermediate layers. (B) The relative linear probing accuracy when using the component from the null space, row space and a random 512-dimensional subspace as component compared to the full component. Both Figures use the MAE-G model.

## A. Discussion: Linear concept directions in ViT MAEs

We have shown that DL is a powerful approach for finding linear concept directions (features) that are strongly correlated with biological concepts such as cell-types and genetic perturbations. From an interpretability perspective, a question that remains, however, is whether these correlations solely appear due to first order effects of complex non-linear structures used by the model to store abstract information, or whether linear directions are actually inherently meaningful to the model? While linear causal interventions offer strong evidence that the latter may indeed at least be partially true for large language models (see e.g.., (Ferrando et al., 2024) for an overview), there exists relatively little evidence for ViT MAEs besides the high linear probing accuracies on e.g., natural and microscopy image classification tasks (Huang et al., 2022; Alkin et al., 2024).

In this section, we provide an argument further supporting the hypothesis that MAEs may rely on linear concept directions when processing data by analyzing at which point in the model are the concepts are the most linearly separable.

**Separation into row- and nullspace.** We note that standard MAE architectures (Huang et al., 2022) use two different embedding dimensions for the encoder block and the decoder block. Both blocks are connected via an *encoder-decoder projection matrix* $W : \mathbb{R}^{d_e \times d_d}$ with, in our case, $d_e = 1664$ (ViT-G model from (Zhai et al., 2022)) and $d_d = 512$. This projection matrix gives raise to a separation of the the tokens into the row-space and null space of $W$, $x = x_{\text{row}} + x_{\text{null}}$ where only the information stored in $x_{\text{row}}$ is passed to the decoder. ViTs and more generally transformer models have shown to align the basis across layers, allowing for decoding of tokens from intermediate layers (Alkin et al., 2024). We visualize this behavior in Figure 7a where we show the reconstructions when using the tokens from intermediate layers. Thus, we observe that the row-space component $x_{\text{row}}^l$ of tokens from early and intermediate layers $x^{(l)}$ already store a reconstruction of the masked image that is refined over the layers. Thus the question appears what is the role of the null space component $x_{null}$ which won't be passed to the decoder and thus serves as a "register" (in analogy to (Darcet et al., 2023))?

**Component-wise linear probing** We analyze in Figure 7b the different components, showing the relative linear probing accuracy of the probing accuracies using the *null* and *row space* components, compared to the entire token (dashed line at 1) across different layers. As observed, the null space component consistently yields the same probing accuracy as the entire token, while the row space component yields significantly lower accuracy. For comparison, we also show the relative probing accuracy when using a random $d_d$-dimensional subspace (the same dimension as the row space), which consistently yields higher accuracy than that obtained from the row space. These findings suggest that biological concepts (i.e., genetic perturbations) are most linearly separable in the component used only for internal processing during the forward pass and not passed to the decoder, and therefore aligns with the hypothesis that the model represents abstract concepts as linear directions accessed by the layers while processing the data (Bricken et al., 2023).

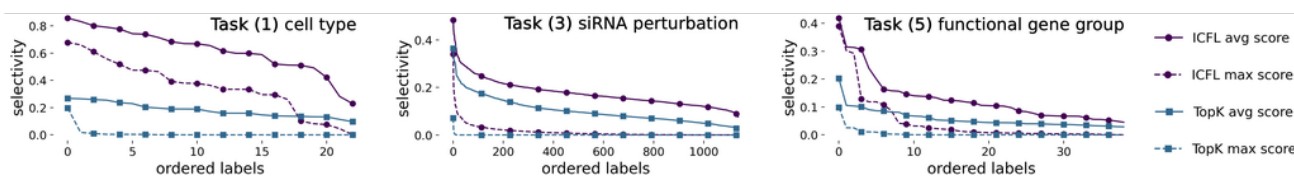

Figure 8. **The highest selectivity scores among all features for each label**, ordered separately for each line starting with maximum score.

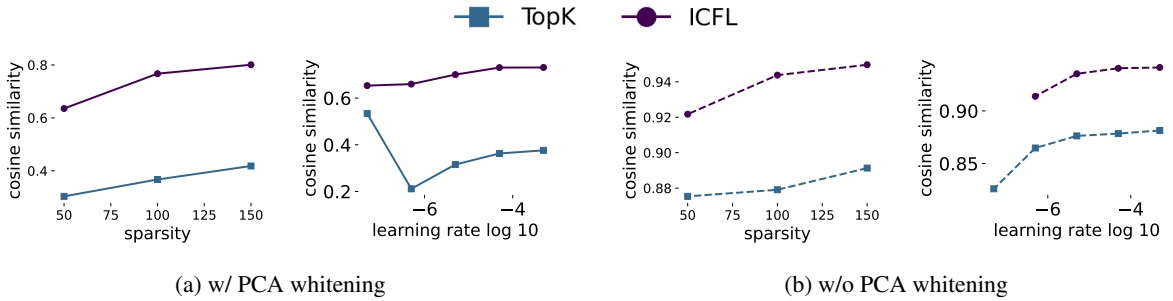

(a) w/ PCA whitening                                      (b) w/o PCA whitening

Figure 9. The cosine similarity between the original tokens and the reconstructed tokens for ICFL and TopK-SAE, **(A)** with PCA whitening and **(B)** without, as a function of the sparsity (first and third), *i.e.* # of non-zeros, and $\log_{10}$ learning rate (second and fourth).

## B. Interpretable features

In this section, we present additional visualizations of crops strongly correlated with selected feature directions. In the spirit of recent works for LLMs (Bricken et al., 2023), we only present a qualitative analysis that aims to highlight non-trivial, complex, and interpretable patterns captured by these features.

For completeness, Figure 11 shows the same crops as Figure 1 but this time all 6 most correlated and anti-correlated crops. We further present in Figures 12 to 16 additional examples similar to Figure 10 for images strongly correlated with different features. In addition to the heat-map and the entire crop, we also plot the patches that are most strongly correlated with the feature. We make two important observations: a) we can see clear interpretable patterns for which patches are most strongly correlated with the cells, posing a promising area for future research on interpreting and validating concept directions found in large foundation models for microscopy image data; b) we see that the most correlated patches are robust to light artifacts, which can be seen best in the last column in Figure 12.

## C. Ablations

In this section we present ablations on type of token, model size, sparsity and learning rate. If not further specified, we always use features extracted from ICFL using PCA whitening.

**Attention block**   It is common in the literature to use representations from the MLP output or the attention output (Bricken et al., 2023; Tamkin et al., 2023; Rajamanoharan et al., 2024a). We compare in Table 4 the test balanced accuracy when taking representations from the residual stream and attention output. We observe that both result in similaraccuracies. We make the same observation in Figure 17a and 17b showing an ablation for the linear probes trained on the reconstruction using the same setting as described in Section 6. Moreover, we compare in Figure 18 the selectivity scores as in Figure 8, confirming further that the residual stream and the attention output show a similar behavior. The only exception is TopK for cell types, where the attention outputs result in significantly better selectivity scores, however, still substantially below the ones obtained by ICFL.

| Residual stream | 97.2% | 87.8% | 51.6% | 94.6% | 32.1% |
|---|---|---|---|---|---|
| Attention output | 96.8% | 85.8% | 52.5% | 94.6% | 32.1% |

Table 4. **The balanced test accuracy (BTA) for representations** taken from the residual stream (BTA row from Table 4) and the attention output.

**Model size** We further investigate the model size, as shown in Figures 17a and 17b, where we compare the linear probes for the MAE-G (referred to as Ph2 with 1.9B parameters) with the much smaller model MAE-L (referred to as Ph1 with 330M parameters). We observe that for simple tasks like classifying cell types, both models yield similar performances. However, we observe consistent improvements on complex classification tasks (3,5), both for the probes trained on the original representations, as well as the reconstructions from ICFL and TopK. This demonstrates that dictionary learning benefits from scaling the model size.

We further plot in Figure 19 the selectivity scores. For ICFL, we consistently observe improvements when increasing the model size, while for TopK SAE, we see a significant drop. Interestingly, this drop does not occur for the probing accuracy on the reconstructions in Figures 17a and 17b. This suggests that, although capturing meaningful signals in the reconstructions, TopK SAE faces more difficulties in finding "interpretable" features with high selectivity scores from richer representations post-processed using PCA whitening.

**Sparsity** As a third ablation, we plot in Figure 9 the cosine similarity of the original tokens and the reconstructed token from the DL for both TopK-SAE and ICFL. We observe that the reconstruction quality of ICFL is much higher than TopK SAE for the same sparsity constraints. This trend persists across all levels of sparsity. The unsupervised reconstruction quality measured by the cosine similarity (or the related $\ell_2$-error) has been often used as a benchmark for SAEs (Rajamanoharan et al., 2024a; Gao et al., 2024).

**Learning rate** As a last ablation, we also plot in Figure 9 the cosine-similarity for different learning rates. Since PCA whitening leads to more dense tokens, we expect that a decrease in the cosine-similarities, which is also the case when comparing the solid lines (w/o PCA whitening) with the dashed lines (w PCA whitening). Except for TopK-SAE with PCA whitening the reconstruction quality slightly increases with the learning rate (likely due to too few training for small learning rates) and flattens after a learning rate of $5 \times 10^{-5}$, which we choose for all experiments in this paper. Moreover, we observe that TopK-SAE experiences high instabilities when combined with PCA whitening, which is not the case for ICFL.

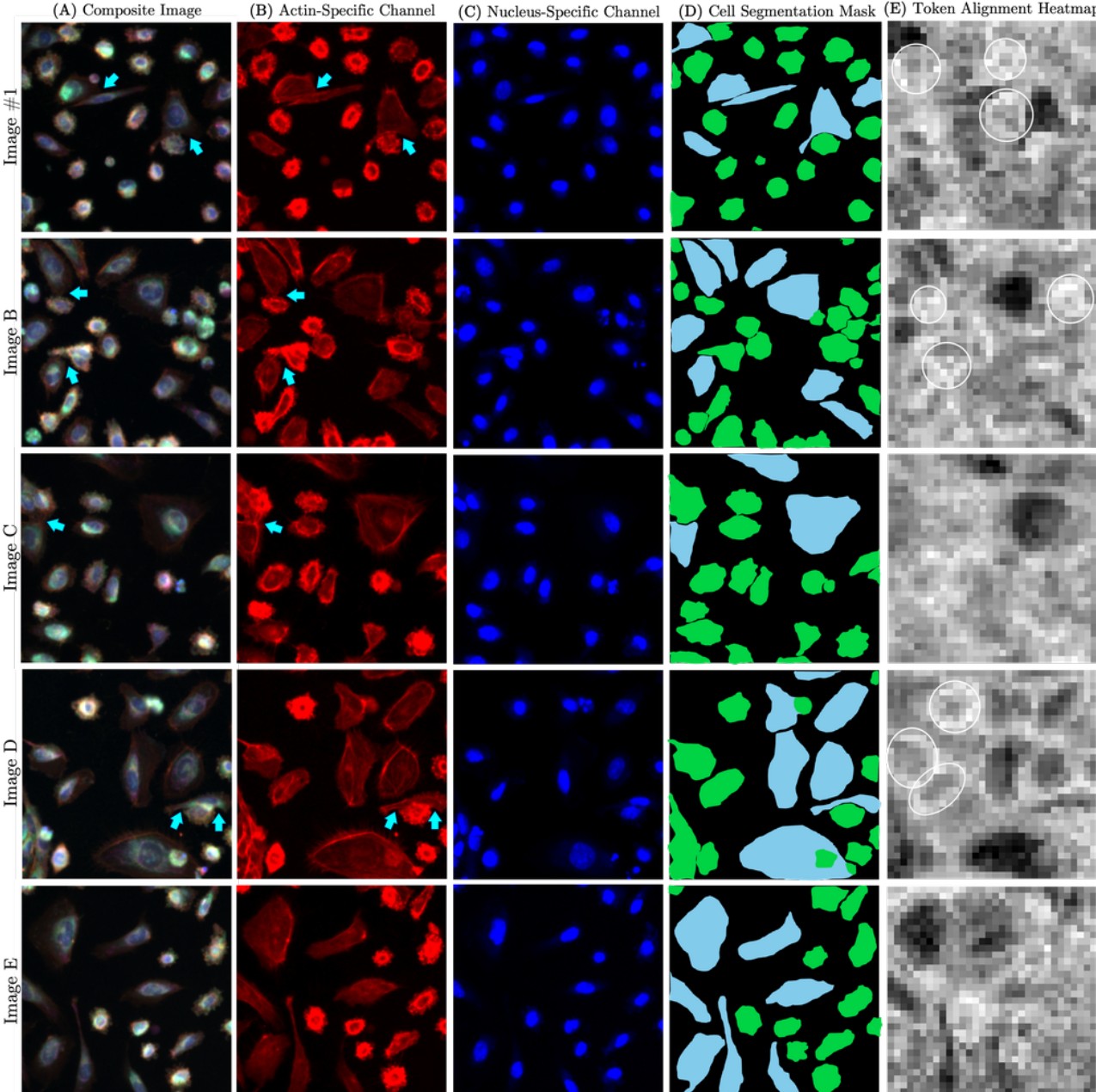

*Figure 10.* **Sample visualizations** of **(A)** composite images and their **(B)** nuclei- and **(C)** actin-staining channels which strongly correlate with a selected feature from a single functional gene group — *adherens junctions*. Plotted by side are the **(D)** cell category-specific single-cell segmentation masks and **(E)** token-level heatmaps of the inner products of the individual tokens with the selected feature direction for 5 out of 8 strongest correlated images per feature direction. Highlighted are correctly perturbed cells (yellow masks) and the cells which most likely remain unperturbed (magenta masks), which are the only instances attempting to establish cell-cell connections (cyan arrows) as they produce the gene to form functional *adherens junctions*.

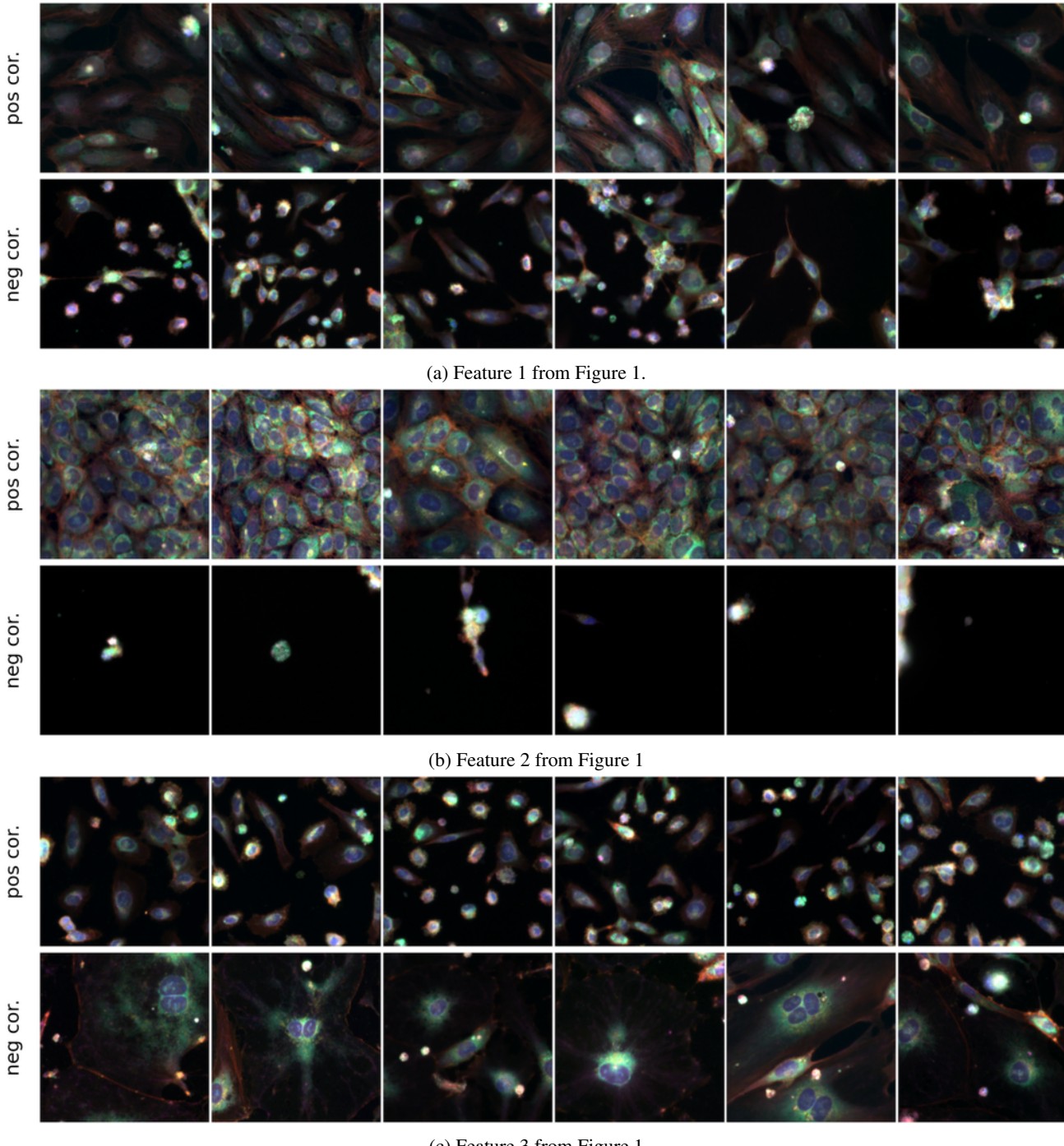

(a) Feature 1 from Figure 1.

(b) Feature 2 from Figure 1

(c) Feature 3 from Figure 1

*Figure 11.* **Additional visualization of images most and least correlated with selected concept directions.** For each row in Figure 1 we also include the crops that are the most correlated with the feature direction in the opposite direction. More precisely, for each feature we show the 6 most positively (first row) and negatively (second row) correlated crops. For each of the three features we observe a clear concept shift along the feature direction (going from negatively correlated to positively correlated).

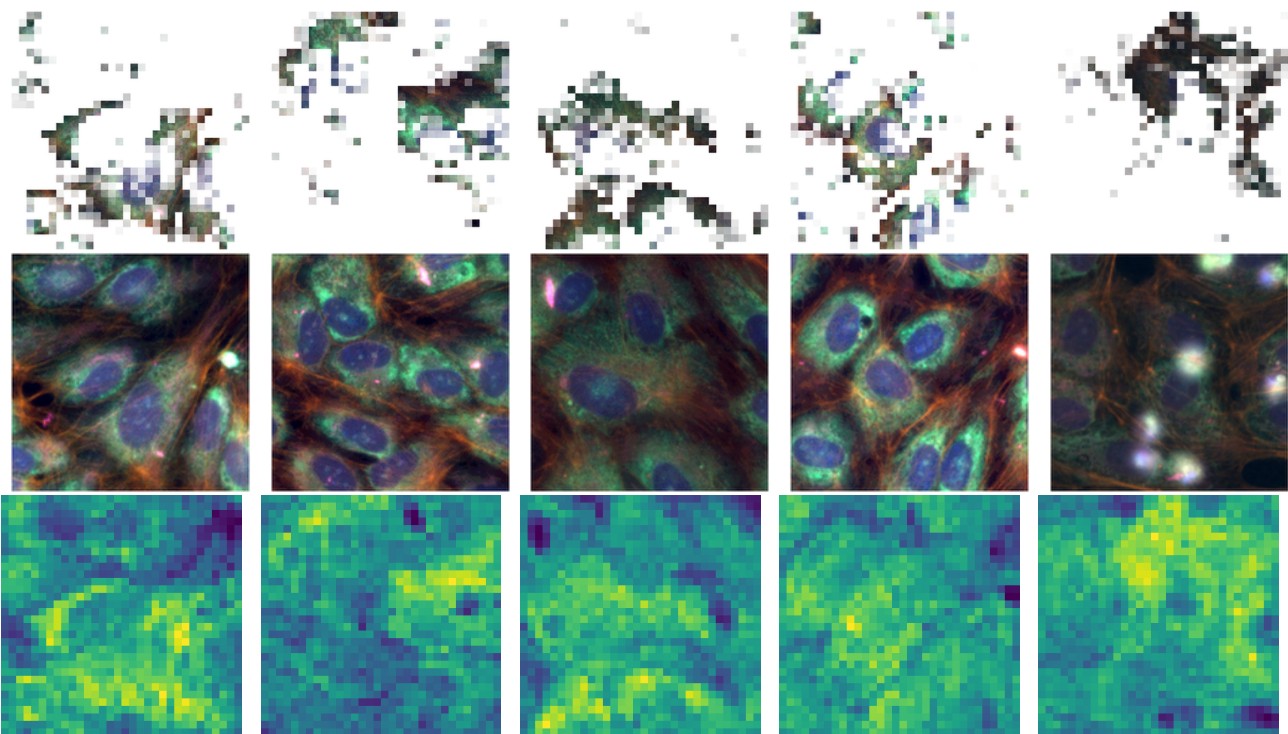

*Figure 12.* **Visualization of sample images aligned with a feature direction with plausible biological interpretation.** This feature appears to be focusing on the endoplasmic reticuli and nucleoli channel (cyan area) surrounding the nucleus. These are expanded relative to the usual morphology of HUVEC cells.

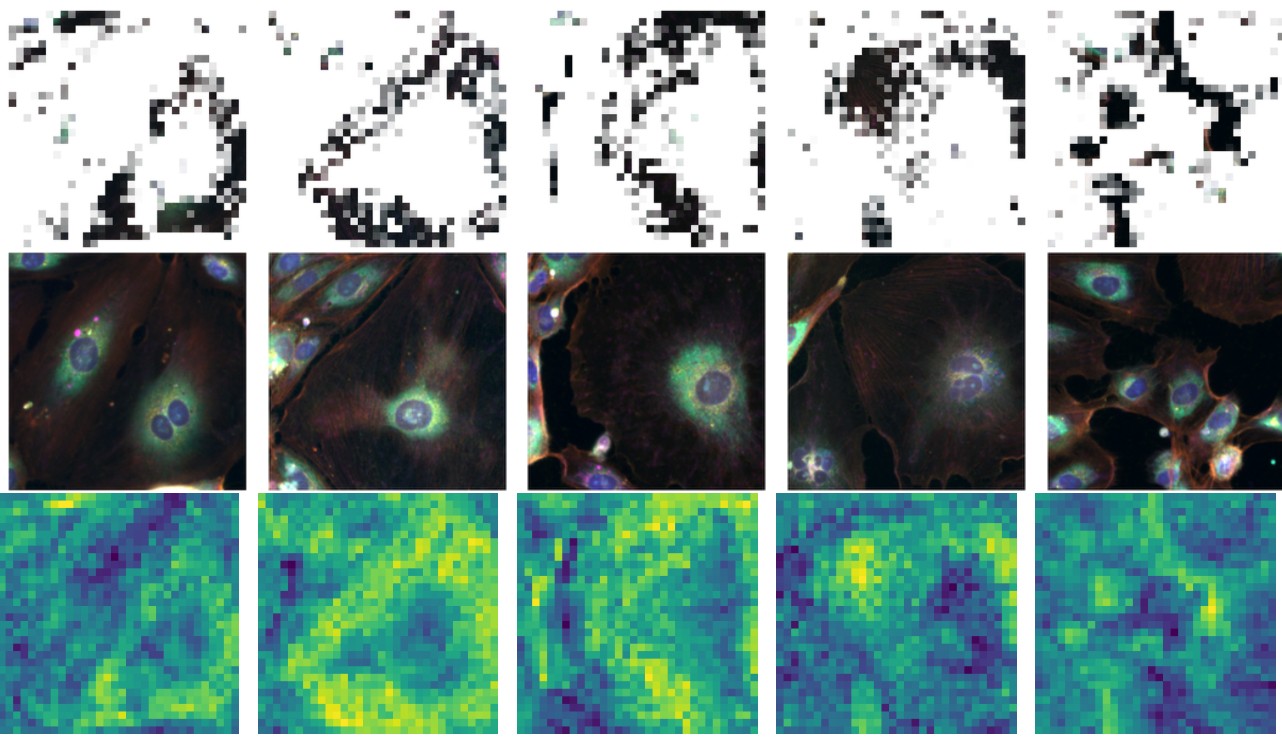

*Figure 13.* **Visualization of sample images aligned with a feature direction with plausible biological interpretation.** This feature appears to be firing for cells that are unusually large with spread out actin. Note that the feature focuses on the actin channel (red) surrounding the cell.

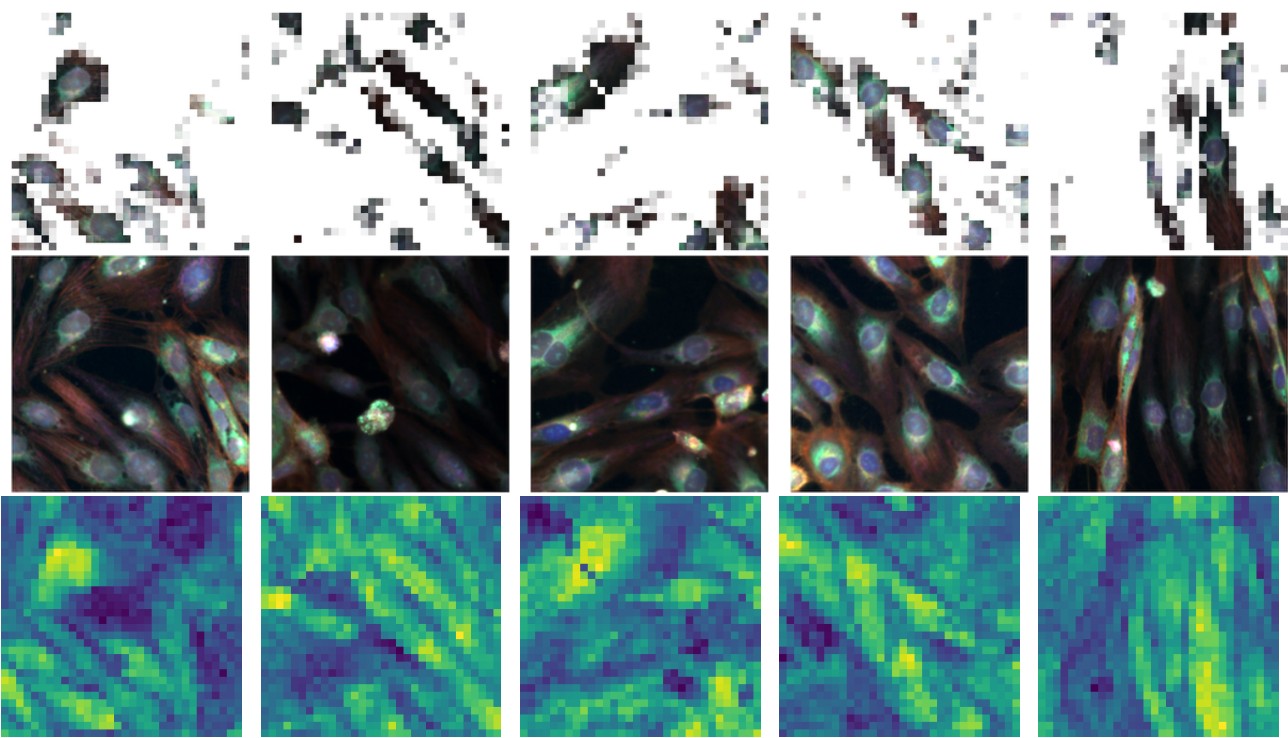

*Figure 14.* **Visualization of sample images aligned with a feature direction with plausible biological interpretation.** This feature appears to be active for long spindly cells, with the features are most aligned for the long "stretched out" section of the cells.

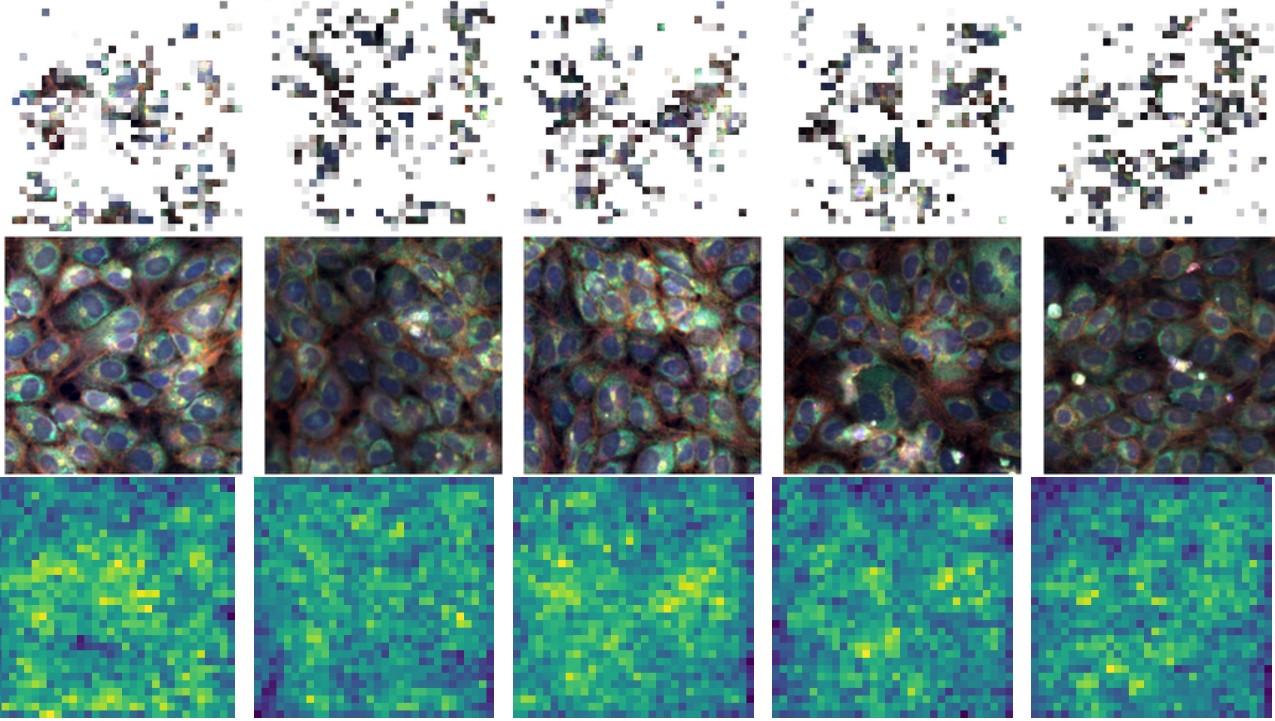

*Figure 15.* **Visualization of sample images aligned with a feature direction with plausible biological interpretation.** This feature is active for tightly clumped cells. The heatmaps are less clearly interpretable for these images, but appear to be active when neighboring nuclei are touching.

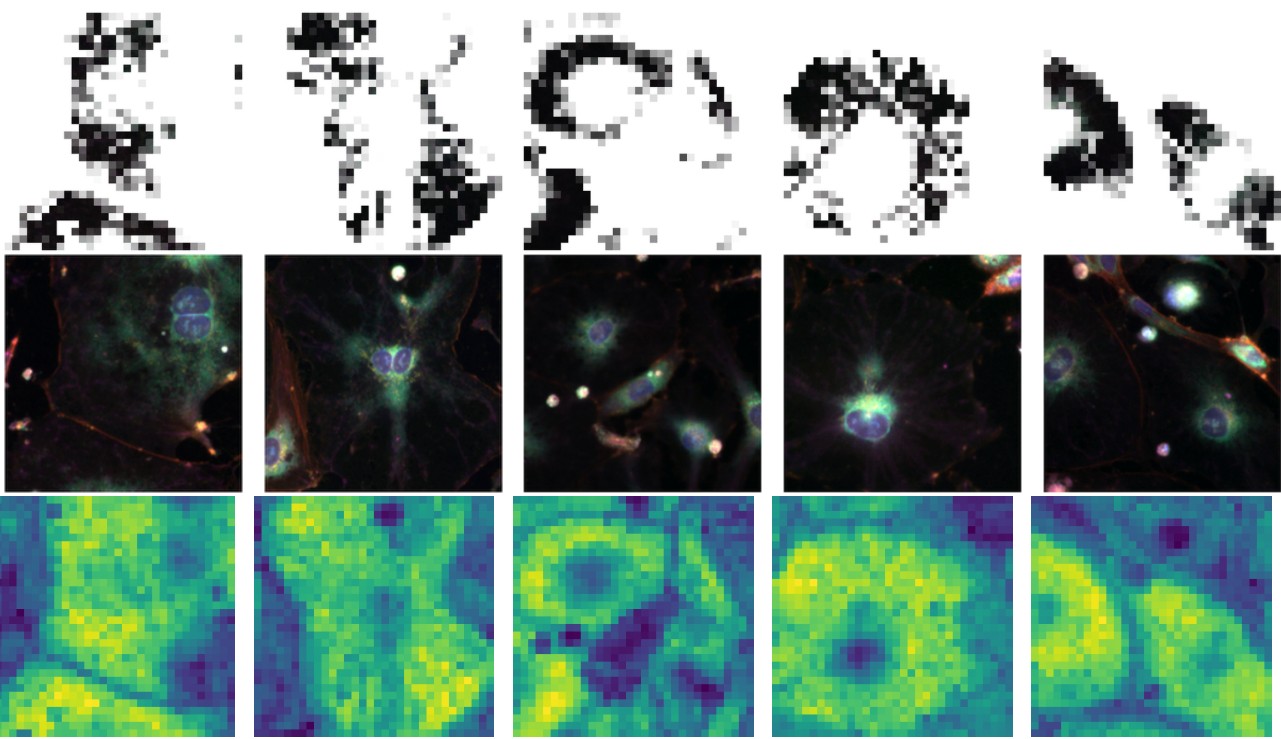

*Figure 16.* **Visualization of sample images aligned with a feature direction with plausible biological interpretation.** This feature shows a similar behavior to the feature in Figure 13

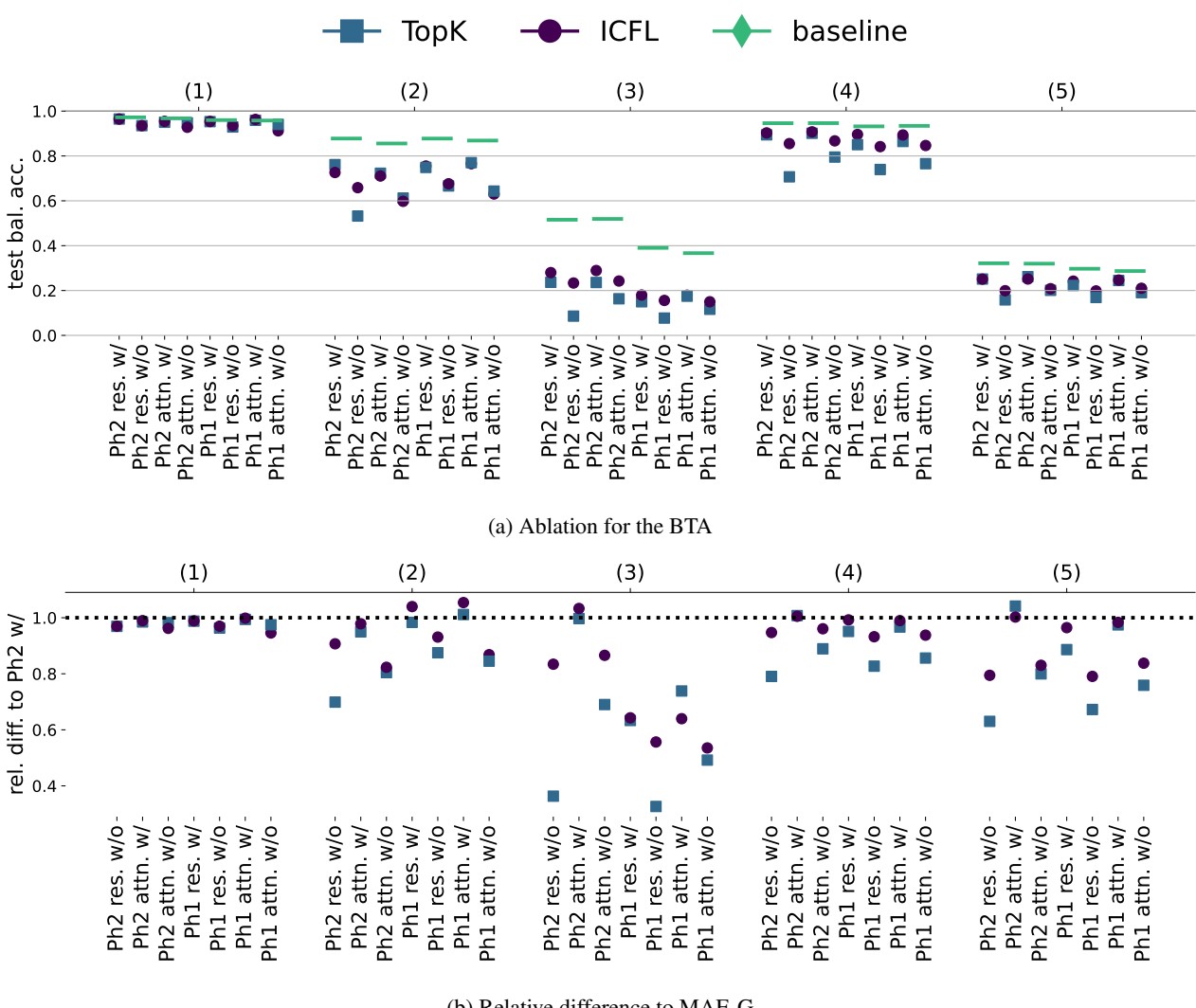

(a) Ablation for the BTA

(b) Relative difference to MAE-G

*Figure 17.* **Comparison of ICFL with TopK SAEs performance. (A)** The BTA of linear probes trained on the original representation (solid lines) and reconstructions from ICFL features and TopK SAEs for representations taken from the residual stream and attention output of *MAE-G* (larger model) and *MAE-L* (smaller model), as well as with PCA whitening and without. **(B)** Same as *(A)* but depicting the relative difference in linear probing accuracy compared to *MAE-G* residual stream using PCA.

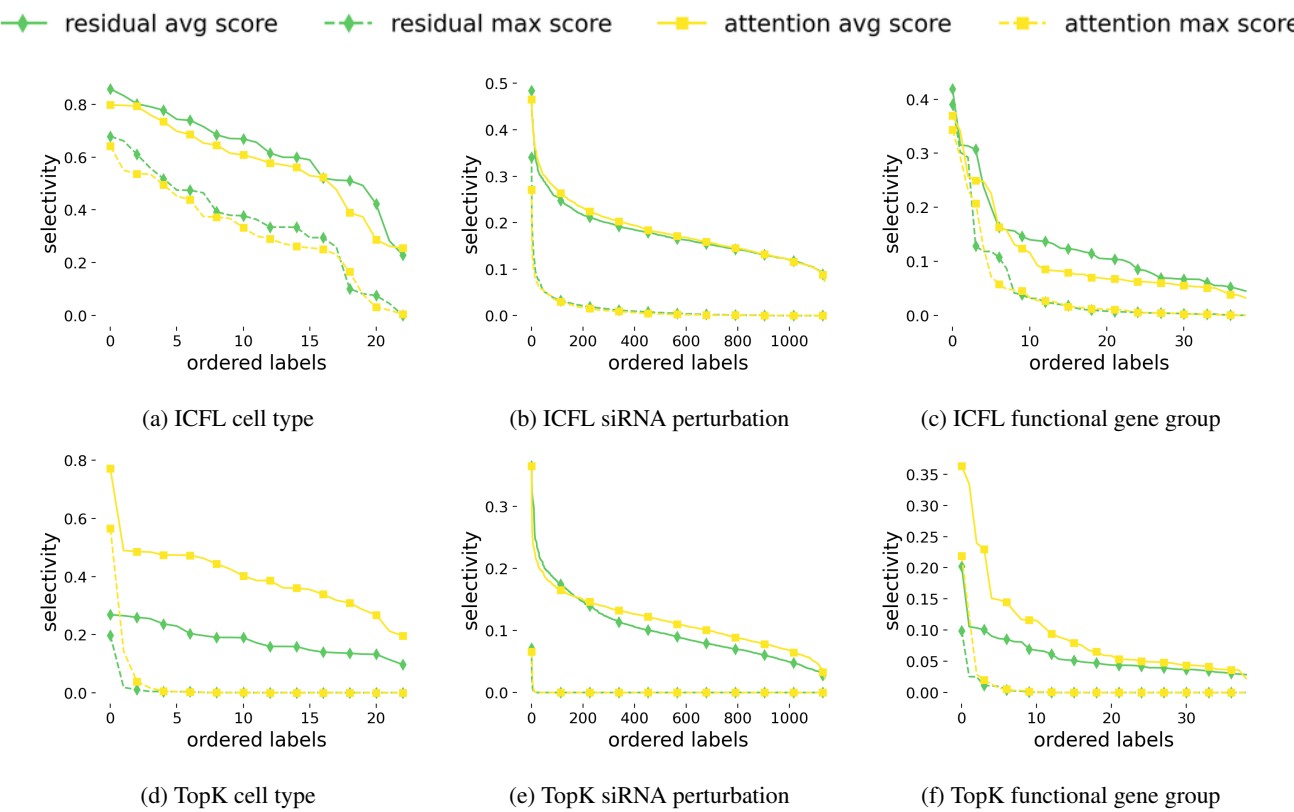

*Figure 18.* **The selectivity scores** as in Figure 8 for ICFL (top row) and TopK (bottom row) when using representations from the residual stream (green) and the attention block (yellow).

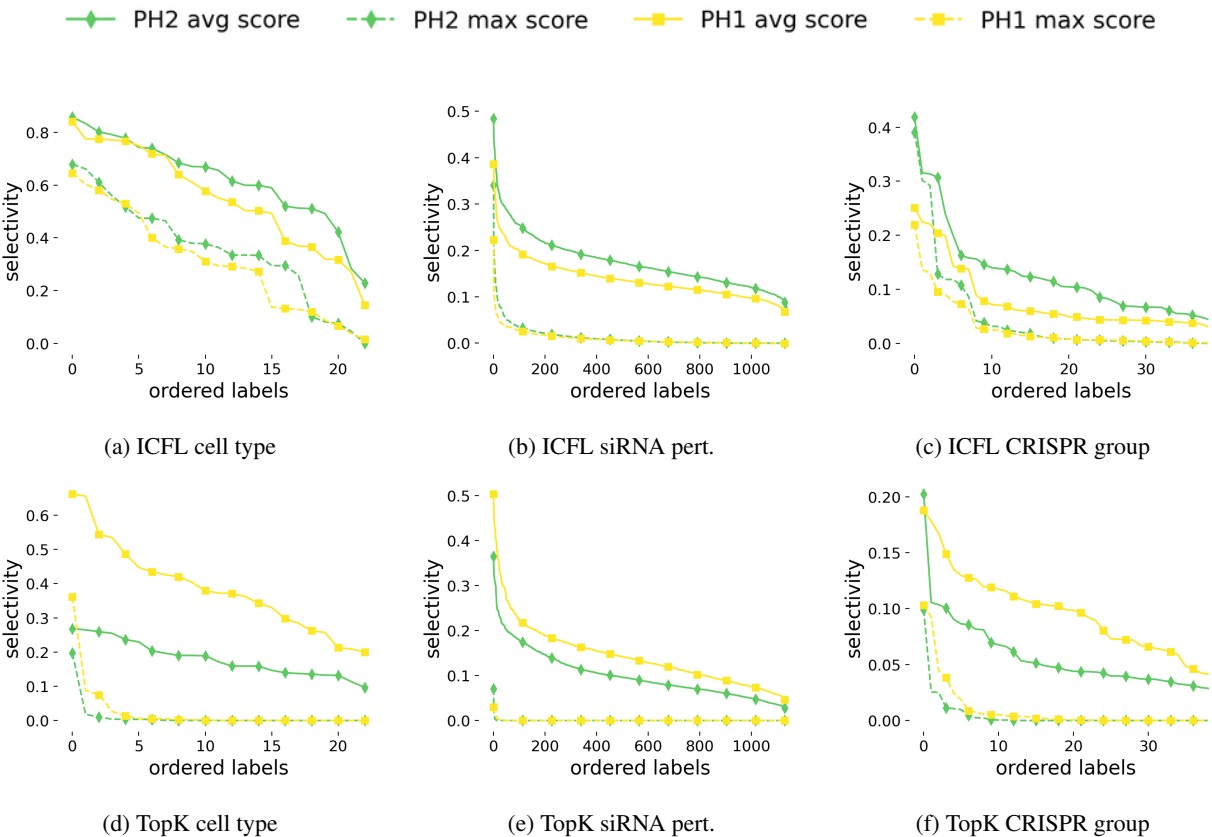

*Figure 19.* **The selectivity scores** as in Figure 8 for ICFL (top row) and TopK (bottom row) when using representations from the residual stream from *MAE-G* (green) and *MAE-L* (yellow) using PCA whitening.

