# OpenReview forum: "Towards scientific discovery with dictionary learning: Extracting biological concepts from microscopy foundation models"
_ICML.cc/2025/Conference — ICML 2025 poster_

### Official Review · Reviewer_e1Nw · 2025-03-08

**Overall Recommendation:** 1

**Summary:**

This paper proposes a new algorithm for dictionary learning, namely Iterative Codebook Feature Learning (ICFL), which can be optionally augmented with the PCA whitening technique. This technique is then applied to interpret features learned in masked autoencoder models trained on microscopy images of cells. The authors present case studies on the extracted features.

**Claims And Evidence:**

The main claims are:
- The proposed approach successfully retrieves biologically meaningful concepts, such as cell type and genetic perturbations.
- The proposed approach can help better understand morphological changes in cells induced by genetic perturbations.

The first claim is supported via experimental results. But some comparisons are not convincing. For example, in Figure 2B, the results of ICFL are so sparse that I do not really know how to compare them against the baseline (CP) performances.

The second claim is based on a specific case study, which presents some evidence but does not adequately convince the readers on the generalizability of the approach.

**Essential References Not Discussed:**

I'm not aware of any missing related works.

**Experimental Designs Or Analyses:**

The experiment **lacks comparisons** between the proposed methods and other existing approaches to interpretability. It is not clear how much practitioners can gain from using ICFL compared with other methods.

**Methods And Evaluation Criteria:**

The evaluation criteria seem sound to me.

**Other Comments Or Suggestions:**

Other minor presentation issues:

- Please check the use of `\citep` and `citet`. There are many mistakes in the paper, e.g., lines 133 (right), 134 (right), 151 (right), 156 (right), 249 (left), etc.

- Line 125 (right): $\\{x\\}_{i=1}^N$ $\to$ $\\{x\_i\\}^N\_{i=1} $.

- Line 184 (left): MP $\to$ OMP.

- Line 205 (right): correspond $\to$ correspond to.

**Other Strengths And Weaknesses:**

## Strengths
- Making scientific discovery with machine learning is an intriguing research topic.

## Weaknesses
- The presentation of the paper, the algorithm part in particular, looks sloppy to me.
  - Abuse of notations. Section 4 uses $W$ and $W_{\text{dec}}$ interchangeably without any additional explanations. I assume that $W_{\text{dec}}$ follows the notations in SAE, but this is inappropriate since there is no $W_{\text{enc}}$. In Equation (4), the use of $W$ and $W_{\text{dec}}$ is not even consistent in a single equation.
  - Missing details. The "computing the features $z$ using batched-OMP" paragraph assumes $W_{\text{dec}}$ is given. However, no where in the main paper is the initialization of $W_{\text{dec}}$ specified.
  - Inconsistent formats. The OMP algorithm is described using bullet points. The batched-OMP is described in (very informal) texts and pseudo code. Making those descriptions consistent would help clarify the modifications in the new method.

- The proposed Iterative Codebook Feature Learning (ICFL) seems a simple extension of existing methods to me.  The PCA whitening is also a known technique. Thus, **the technical contribution of the paper is limited**. While I value the simplicity in the design of ML algorithms, the authors need to present very solid comparisons against existing methods, show the generality on more foundation model families, and discuss insights on why the simple method is favorable.

**Questions For Authors:**

- In line 163 (right), are $W_i$ and $x^{(t)}$ both column vectors? How is $|W_ix^{(t)}|$ defined?

- Are the MAE models open-sourced? What are the training configurations of the models? Those details should have been documented in the paper.

**Relation To Broader Scientific Literature:**

From the interpretability perspective, this paper introduces a new algorithm based on dictionary learning.

**Theoretical Claims:**

No theorem in this paper.

---

> ### Author Rebuttal · Authors · 2025-03-31
>
> We thank the reviewer for their constructive review. We would like to respond below to a few comments made by the reviewer.
>
> > The experiment lacks comparisons between the proposed methods and other existing approaches to interpretability. It is not clear how much practitioners can gain from using ICFL compared with other methods.
>
> While a more extensive benchmark evaluation would indeed be valuable, we would like to stress that the emphasis of this paper lies in the use of dictionary learning / SAEs in the context of biological representation learning. In particular, we aim to explore how ideas developed in mechanistic interpretability for LLMs could potentially be used in the future for scientific discovery. A benchmark comparison therefore goes beyond the scope of this paper.
>
> >  show the generality on more foundation model families
>
> In this paper, we prioritize an interdisciplinary approach and an in-depth analysis of features extracted from one specific model rather than multiple models. This is a common approach in the existing literature (see, e.g., [1,2]) for two reasons.
> - First, assessing the quality of extracted features is a highly domain-specific task, hence our usage of a SOTA model for cell microscopy data, rather than a generic vision foundation model. An attempt to apply the method to more foundation models, would have necessitated a less detailed study of the features that are learned.
> - Second, we note that the overwhelming majority of prior work [e.g. 1,2] on dictionary learning techniques have also only been applied to a single modality (typically text) with a single model. This approach is common for the reason outlined above (in any domain, careful analysis of the features requires extensive domain expertise), and access to large scale foundation models, the corresponding datasets, and the necessary infrastructure to run these experiments is often limited.
>
> > The presentation of the paper, the algorithm part in particular, looks sloppy to me.
>
> We thank the reviewer for making us aware of the inconsistencies  in the notation. We switched between W and Wdec to better highlight the similarity between ICFL and TopK SAEs, however, there are indeed some minor inconsistencies due to smaller edits before submitting the paper, which we will resolve. Moreover, Wdec is initialized using the standard pytorch initialization for a linear transform (nn.Linear). We will mention this in the paper.
>
> > In line 163 (right), are
>
> It should indeed be W_i^T, thank you for noticing this typo.
>
> > Are the MAE models open-sourced? What are the training configurations of the models? Those details should have been documented in the paper.
>
> Unfortunately, we are not able to open source the MAEs at the time of writing this response, but they are based on the standard implementation of the MAEs described in [3]. Moreover, the training details for the MAEs are as described in [4].
>
>
> [1] Gao, Leo, et al. "Scaling and evaluating sparse autoencoders." arXiv preprint arXiv:2406.04093 (2024).
>
> [2] Bricken, Trenton, et al. "Towards monosemanticity: Decomposing language models with dictionary learning." Transformer Circuits Thread 2 (2023).
>
> [3] He, Kaiming, et al. "Masked autoencoders are scalable vision learners." Proceedings of the IEEE/CVF conference on computer vision and pattern recognition. 2022.
>
> [4] Kraus, Oren, et al. "Masked autoencoders for microscopy are scalable learners of cellular biology." Proceedings of the IEEE/CVF Conference on Computer Vision and Pattern Recognition. 2024.

---

> > ### Comment · Reviewer_e1Nw · 2025-04-02
> >
> > I thank the authors for the rebuttal and I appreciate the opportunity to review this interdisciplinary work.
> >
> > While I appreciate the authors' efforts, I find that the machine learning methodology lacks sufficient depth and novelty from a technical perspective. As my expertise lies primarily in machine learning rather than computational biology, I defer to other reviewers and AC with domain expertise for assessment of the biological findings.
> >
> > From my perspective as a machine learning researcher, the paper does not provide substantial new methodological insights that would advance the field. Therefore, I maintain my original rating.

---

> > > ### Author Response · Authors · 2025-04-04
> > >
> > > We are unlikely to reach agreement about depth or novelty - *we view the simplicity of the method as a key strength as simple methods tend to have higher impact; for example, the same “lacks sufficient [technical] depth and novelty” comment could be applied to Brickle et al. [2023] who “just” used a sparse autoencoder (a well-known technique), but have had a significant impact*.
> > >
> > > Instead, we ask the reviewer to reconsider their position in light of the official ICML reviewer guidelines for 2025 which explicitly encourage taking a broad view on originality in order to avoid inherently subjective debates about perceived novelty. In particular, the guidelines state,
> > >
> > > > We **encourage you to be open-minded** in terms of potential strengths. For example, **originality may arise from creative combinations of existing ideas**, removing restrictive assumptions from prior theoretical results, or **application to a real-world use case** (particularly for application-driven ML papers, indicated in the flag above and described in the Reviewer Instructions). [emphasis added]
> > >
> > > We present a method that:
> > > - combines **matching pursuit** with a **learned dictionary** that it is optimized by gradient descent and a **PCA preprocessing step**. Each of these three ideas is known, but we provide extensive experiments to demonstrate that it is their combination that leads to strong performance in a setting not previously studied (dictionary learning for image-only models that receive no text supervision); and
> > > - we provide an in-depth evaluation on a real world use case.
> > >
> > > Both of these points are supported with detailed experiments and ablations that show both the effectiveness of dictionary learning in general on the real-world use case, and the effectiveness of the novel combination of existing ideas in improving over the standard technique used in this area (Top-K SAEs).

---

### Official Review · Reviewer_Sady · 2025-03-14

**Overall Recommendation:** 3

**Summary:**

This paper explores the application of dictionary learning (DL) to extract biologically meaningful concepts from large-scale masked autoencoders (MAEs) trained on microscopy images. The authors introduce Iterative Codebook Feature Learning (ICFL), a dictionary learning algorithm adapted from the Matching Pursuit (MP) algorithm, combined with PCA whitening on a control dataset. The approach aims to uncover sparse, interpretable features from cell imaging data that correspond to biological concepts such as cell types and genetic perturbations. The authors validate their method through classification tasks, interpretability analyses, and comparisons with Top-K sparse autoencoders (TopK SAEs) and handcrafted features from CellProfiler (CP). They show that ICFL improves selectivity, reduces “dead features,” and retains biologically relevant signals. The results suggest that dictionary learning can be effectively applied to bioimaging data, potentially enabling new insights in drug discovery and genetic perturbation analysis.

**Claims And Evidence:**

1. Lack of ablation experiments for sparse representations -- The paper emphasizes that sparse dictionary learning is essential for interpretability, but does not conduct a sparse vs. non-sparse comparison experiment. The lack of such an ablation experiment affects the credibility of the conclusion.
2. Lack of hyper-parameter sensitivity analysis -- ICFL relies on several hyper-parameters (e.g., sparsity, number of iterations, PCA whitening strength), but the paper fails to conduct sensitivity analysis on these parameters, which affects the assessment of model robustness.
3. Single way of assessing feature interpretation - the paper mainly uses Linear Probing to assess whether the features are biologically relevant, but does not use feature clustering analysis (whether biologically meaningful clusters can be formed spontaneously).
The use of these additional assessments can improve the persuasiveness of the paper.

**Essential References Not Discussed:**

None

**Experimental Designs Or Analyses:**

The experiments in this paper are mainly conducted on 100,000-level microscope data, and the lack of computational complexity analysis of million-level bio-image data affects the applicability of this method on large-scale datasets.

**Methods And Evaluation Criteria:**

The paper uses PCA whitening + sparse dictionary learning to extract features, but does not compare it with broader explanatory methods (e.g., topological data analysis, decomposable representation learning), which affects the full assessment of its methodological uniqueness.

**Other Comments Or Suggestions:**

It is recommended that the ICFL computational complexity analysis be supplemented with operational efficiency tests on large-scale data to enhance the utility of the method.

**Other Strengths And Weaknesses:**

None

**Questions For Authors:**

1. What is the computational complexity of ICFL on millions of microscope data? Has it been tested for efficiency on large-scale data?
2. can the extracted dictionary features be matched with known biomarkers?
3. does PCA whitening result in loss of certain biologically relevant information? Are there ablation experiments?
4. have alternative methods of feature selection been considered (e.g., feature extraction based on topological data analysis)?

**Relation To Broader Scientific Literature:**

None

**Theoretical Claims:**

ICFL relies on sparse representations to improve interpretability, but does not provide ablation experiments to demonstrate the necessity of sparsity for biological feature extraction.

---

> ### Author Rebuttal · Authors · 2025-03-31
>
> We thank the reviewer for their detailed review and for raising helpful questions. We would like to respond below to a few comments made by the reviewer.
>
> > Single way of assessing feature interpretation
>
> While linear probing is one important way to evaluate the quality of the features, we disagree with the reviewer’s comment that the paper does not use feature clustering analysis. In Figure 4 we show examples for how well clusters are separated along selected feature directions. Moreover, the selectivity scores (as displayed in Figure 2) essentially measures how well features can separate individual genetic perturbations from the average of all perturbations. May we ask the reviewer whether this answers the reviewers concern?
>
>
> > Lack of hyper-parameter sensitivity analysis… sparsity, number of iterations, PCA whitening strength
>
> We agree that careful ablation studies are an important part of a paper. While the computational cost of these experiments prevents covering the full space of hyperparameters (the dictionary learning methods are trained over 40m tokes), we did include the hyperparameters that you suggested (and many more) in the paper:
> - We ablate over the sparsity (Figures 3 and 8), model size of the MAE (Figure 17), type of representation (Figures 17 and 18) and learning rate (Figure 8).
> - We did not find the method to be sensitive to the number of iterations as we trained to approximate convergence.
> - There is no notion of “whitening strength” - the representations are whitened such that they have unit covariance and zero mean on the control cells
>
>
> > Lack of ablation experiments for sparse representations
> > ICFL relies on sparse representations to improve interpretability, but does not provide ablation experiments to demonstrate the necessity of sparsity for biological feature extraction.
>
> The reviewer raises an interesting point. It is per se not clear why one should extract sparse representations instead of dense representations. First, to answer why not dense representations? We recall that the dimension of z (i.e. 8192) is much higher than the dimension of x (i.e. 1664). Without the sparsity constraint, there are infinitely many ways to implement the identity function (that is, Wz = x), and thus the problem is inherently ill-posed. As a result, there is also no reason why one should believe that the learned features should capture biologically meaningful signals.
>
> However, why sparse features? The motivation stems from the superposition hypothesis, which is described in Section 3. While this hypothesis clearly does not hold universally, the question remains - to what extent do it hold? With this paper, we provide at least some evidence that a weaker form of a superposition hypothesis may indeed hold.
>
>
> > but does not compare it with broader explanatory methods (e.g., topological data analysis, decomposable representation learning)
>
> While we agree that broader comparisons are generally useful; comparing across different classes of interpretability methods is very challenging because each class of methods has a different notion of interpretability. As a result, there are not clean quantitative comparisons across interpretability methods that can be made. That said, we believe that a well-designed user study that evaluated these methods for scientific discovery would be extremely valuable, but this goes well beyond the scope of this paper.
>
>
> > The experiments in this paper are mainly conducted on 100,000-level microscope data, and the lack of computational complexity analysis of million-level bio-image data affects the applicability of this method on large-scale datasets.
>
> We believe there is a misunderstanding. We use a subset of the data for evaluation when computing the linear probes. However, to train the dictionaries W, we use 40M tokens (that is, 40M crops). Moreover, we train for 300k steps with a batch size of 8192 tokens.
>
> > 1. What is the computational complexity of ICFL on millions of microscope data? Has it been tested for efficiency on large-scale data?
>
> We did not conduct an extensive run-time analysis given that the cost of compute is negligible compared to the cost of training and fine-tuning the MAEs
> .
> > 2. Can the extracted dictionary features be matched with known biomarkers?
>
> The selectivity score analysis suggests that the extracted features align well with (some) genetic perturbations. May we ask the reviewer to be more precise what sort of experiment they had in mind?
>
> > 3. Does PCA whitening result in loss of certain biologically relevant information? Are there ablation experiments?
>
> The contrary is true, PCA whitening helps to increase the biological signal in the extracted features. We would like to refer the reviewer to Figure 3.
>
> > 4. Have alternative methods of feature selection been considered...
>
> Not at this time, as such methods go beyond the scope of this paper, but we do regard such an analysis as an interesting and important future work.

---

### Official Review · Reviewer_Qi4i · 2025-03-14

**Overall Recommendation:** 5

**Summary:**

This paper adapts techniques from the mechanistic interpretability literature to address the problem of discovering what concepts are learned by foundation models trained on microscopy data. Specifically, they develop a new dictionary learning method, which they call ICFL, based orthogonal matching pursuit to extract factors from a masked autoencoder trained on the Cell Painting dataset. A series of auxiliary probe tasks are used to evaluate the quality of the learned atoms. The features are compared with a manual feature extraction pipeline called CellProfiler.  Finally, the usefulness of the learned concepts is illustrated through an extended case study of the a genetic perturbation experiment.

**Claims And Evidence:**

I found the claims to be direct and well-supported. For example, it is easy to understand what dead features in SAE are, and the results of Table 1 seem to resolve any doubts that their approach suffers from this problem less.  Similarly, Figure 4 convinced me that the representations were of practical value in perturbation experiments.

I was impressed by the depth of the case study. The identification of factors related to adherens junctions struck me as the sort of result that would be of genuine interest to computational biologists.

**Essential References Not Discussed:**

These are not necessarily essential references, but the idea of identifying the major factors of variation in a control pool struck me as quite close to the ideas in:

https://doi.org/10.1038/s41587-024-02463-1
https://doi.org/10.1093/biostatistics/kxv026

**Experimental Designs Or Analyses:**

The ablation experiments in the appendix are helpful for understanding the general properties of IFCL. Despite the comparison with Top-K SAE, some readers might find the lack of more detailed quantitative comparisons (on other datasets or against other baselines) potentially concerning. However, I found the novelty of the application and the qualitative finding from the case study outweighed any concern about precise quantitative benchmarking.

**Methods And Evaluation Criteria:**

The idea of applying SAE to microscopy images from perturbation experiments is well-justified and the proposed probes are appropriate for evaluation. The method of subtracting variation present in control samples is well-motivated by the experimental design.

The only step in the method that I found unusual was how the feature directions are learned. Unless I am misunderstanding something, the objective (4) has a closed form solution for W and b. Using gradient descent with random resets struck me as an unecessarily complicated in an otherwise elegant approach.

**Other Comments Or Suggestions:**

I have no additional comments.

**Other Strengths And Weaknesses:**

The paper is written in a direct and precise style.

**Questions For Authors:**

I have no additional questions.

**Relation To Broader Scientific Literature:**

This work brings together ideas from two cutting-edge areas -- genetic perturbation experiments and mechanistic interpretability. Better methodology for analyzing perturbation experiments is an area of active research.  Researchers have worked with both manual feature extraction and deep learning models for the associated microscopy images, but combining the interpretability/control of manual feature methods with the richness of self-supervised learning strikes me as potentially very impactful.

In the interpretability literature, the IFCL algorithm is closely related to Top-K SAE and OMP, and the authors appropriately represent their contribution.  From an interpretability perspective, much of the novelty is in the application domain. Many interpretability techniques have been understood on LLM embeddings, but few (none?) have successfully demonstrated utility on more advanced scientific imaging data.

**Theoretical Claims:**

There are no theoretical claims.

---

> ### Author Rebuttal · Authors · 2025-03-31
>
> We thank the reviewer for their comments and appreciate their positive feedback on the paper. We were especially pleased to read that the reviewer appreciated our extensive case study. We would like to respond below to a few comments made by the reviewer.
>
> > The only step in the method that I found unusual was how the feature directions are learned. Unless I am misunderstanding something, the objective (4) has a closed form solution for W and b.
>
> We thank the reviewer for raising this point and would like to clarify: Given z, there is indeed a closed-form solution for W and b. However, there are two key reasons why W should not simply be updated to minimize Equation (4):
> First, given that we use more than 40M training tokens (samples), it would be infeasible to optimize (4) directly over all samples. Instead, we rely on mini-batches, making local updates such as gradient descent a natural choice.
>
> Second, the objective in Equation (4) is convex in W and b only if z is fixed. However, z depends on W since it is computed as the solution to Algorithm 1. This raises the question: what is a practical way to update W? Directly solving Equation (4) may significantly alter W without accounting for its dependency on z. This issue is mitigated when we take only a gradient step with respect to the objective in (4) before updating z again. We refer the reader to [1] for a similar algorithm, where the decoder (or dictionary matrix) is also learned via gradient descent.
>
> > Using gradient descent with random resets struck me as an unecessarily complicated in an otherwise elegant approach.
>
> One could think of the random resets as playing a role analogous to the projection step in projected gradient descent, in order to enforce the constraint that cosim(w_i, w_j) < 1-\delta. However, unlike projected gradient, we are not deterministically projecting onto the boundary of the constraint set; instead, the random reset ensures that we are at some (random) point in the interior of the constraint set (with high probability). This trades off the compute costs and algorithmic complexity of a true projection step, for a cheap alternative that we find works well in practice. We will add this discussion to the updated paper.
>
>
>
>
> > These are not necessarily essential references, but the idea of identifying the major factors of variation in a control pool struck me as quite close to the ideas in: [reference to a paper by Zhang et al., 2024, Nat Biotech].
>
> Thank you  for highlighting this interesting work and agree that discussing this line of research is a valuable addition to the paper.. We will extend our related work section accordingly.
>
> [1] Arora, Sanjeev, et al. "Simple, efficient, and neural algorithms for sparse coding." Conference on learning theory. PMLR, 2015.

---

> > ### Comment · Reviewer_Qi4i · 2025-04-05
> >
> > The reference to Arora et al. (2015) and their "neurally plausible algorithm" resolves my concerns about the use of gradient updates in Equation (4). The random reset proposal still seems heuristic -- the projected gradient idea is suggestive but could perhaps benefit from formal study.
> >
> > Nonetheless, I remain enthusiastic in my support of the paper. I feel that the analysis deserves high visibility, because it exemplifies careful thinking about interpretability in a challenging application domain.

---

### Decision · Program_Chairs · 2025-05-01

**Decision:**

Accept (poster)

**Comment:**

The review are split on this paper. On one side, all reviewers agree that the paper propose and innovative application of interpretability for microscopy data, with some interesting results. But one reviewer points out that the methodology is not new and it is unclear that it would lead to good results on another task, limiting the scope of the paper for the broad audience of ICML.

I find that the paper is well written, with a clear exposition of both the methods and the application. The method is also simple and provides some insights on what the large model encodes in their intermediate layers, and the results on the microscopy are intriguing, even though it is not completely clear if they could really lead to scientific discovery (the results right now demonstrate that the model learns some known properties, but it is much harder to find novel ones from the learned representations).
On the other side, it is not completely clear what the novelty is in ICFL (it looks like classic DL with OMP sparse code and gradient update of the dictionary), and it is unclear whether it would lead to better results on other modalities. Moreover, no discussion is given about the runtime.

Overall, I lean mostly toward accepting the paper due to its novelty for the application, which shows a good example of scientific investigation in the results of Large models, with an original contribution. I would still recommend adding a discussion about runtime for both methods and fixing figure 2B to have more points for ICFL.


**Extra remarks:**

- Using OMP for dictionary learning is far from new, and the random reset of unused atoms is a classical step in DL literature. OMP is typically one of the algorithm available in [scikit learn](https://scikit-learn.org/stable/modules/generated/sklearn.decomposition.DictionaryLearning.html#dictionarylearning) and the code implement a random reset of unused atoms.
- Side note on the first answer to **Qi4i**: the second point is not really valid. There is a large body of dictionary learning literature that uses alternate minimization, which alternates between $z$ and $W, b$, and there is a clear justification for this. But I agree that the first point is valid and requires learning with an iterative method, which can be one step of gradient descent.
- l.170 - It is confusing that $k$ is both the number of features selected at each step and the predefined sparsity. I would advise changing it in the algorithm to select $l$ features?
- A reference about dictionary learning with cell imaging that could be interesting to include: Yellin, F., Haeffele, B. D. & Vidal, R. Blood cell detection and counting in holographic lens-free imaging by convolutional sparse dictionary learning and coding. in IEEE International Symposium on Biomedical Imaging (ISBI) (Melbourne, Australia, 2017).
- l.331.c2 - typo: "the by"
- Fig5-title : "actin"->action?